# New light shed on the early evolution of limb-bone growth plate and bone marrow

**Jordi Estefa[1]\*, Paul Tafforeau[2], Alice M Clement[3], Jozef Klembara[4], Grzegorz Niedźwiedzki[1], Camille Berruyer[2], Sophie Sanchez[1,2]\***

[1]Department of Organismal Biology, Evolution and Development, Uppsala University, Uppsala, Sweden; [2]European Synchrotron Radiation Facility, Grenoble, France; [3]Flinders University, College of Science and Engineering, Adelaide, Australia; [4]Comenius University in Bratislava, Faculty of Natural Sciences, Department of Ecology, Bratislava, Slovakia

**Abstract** The production of blood cells (haematopoiesis) occurs in the limb bones of most tetrapods but is absent in the fin bones of ray-finned fish. When did long bones start producing blood cells? Recent hypotheses suggested that haematopoiesis migrated into long bones prior to the water-to-land transition and protected newly-produced blood cells from harsher environmental conditions. However, little fossil evidence to support these hypotheses has been provided so far. Observations of the humeral microarchitecture of stem-tetrapods, batrachians, and amniotes were performed using classical sectioning and three-dimensional synchrotron virtual histology. They show that Permian tetrapods seem to be among the first to exhibit a centralised marrow organisation, which allows haematopoiesis as in extant amniotes. Not only does our study demonstrate that long-bone haematopoiesis was probably not an exaptation to the water-to-land transition but it sheds light on the early evolution of limb-bone development and the sequence of bone-marrow functional acquisitions.

**\*For correspondence:**
jordi.estefa@gmail.com (JE);
sophie.sanchez@ebc.uu.se (SS)

**Competing interests:** The authors declare that no competing interests exist.

## Introduction

Tetrapod long bones are among the most studied skeletal elements in the field of bone biology as they constitute a unit of reference for understanding the development and biomechanics of the appendicular skeleton (e.g. *Duboule, 1994*; *Fröbisch, 2008*; *Hall, 2008*; *Shubin et al., 1997*). The recent discovery of fossil tetrapod trackways (*Ahlberg, 2018*; *Niedźwiedzki et al., 2010*) suggested that limbs supported weight and engaged substrate locomotion earlier than previously thought in early tetrapod evolution. Not only crucial for their biomechanical properties, long bones also host bone marrow including stem-cell niches for the production of blood cells, that is haematopoiesis (*Orkin and Zon, 2008*). After birth, bone marrow is the definitive haematopoietic system in mostly terrestrial mammals and many other tetrapods (*Akiyoshi and Inoue, 2012*; *Kapp et al., 2018*; *Orkin and Zon, 2008*) but not in fish or some aquatic tetrapods (*Akiyoshi and Inoue, 2012*; *Avagyan and Zon, 2016*; *Kapp et al., 2018*). Indeed, red blood cells are produced in the supraspinal organ in the lamprey, the kidney and liver in actinopterygians (ray-finned fish) and some amphibians (tadpoles and aquatic adults, *Akiyoshi and Inoue, 2012*), and the kidney in lungfish (*Amemiya et al., 2007*; *Kapp et al., 2018*). Several studies proposed that the skeleton would have played a major role in hosting blood-cell production over the water-to-land transition and (1) protecting it against temperature changes (*Weiss and Wislocki, 1956*), (2) protecting it against potential DNA mutations induced by UV exposure on land (*Horton, 1980*; *Kapp et al., 2018*) or (3) providing a better efficiency in red-blood-cell production necessary for metabolically-demanding terrestrial locomotion and aerial respiration (*Tanaka, 1976*). Our study focusses on characterising the

**eLife digest** For many aquatic creatures, the red blood cells that rush through their bodies are created in organs such as the liver or the kidney. In most land vertebrates however, blood-cell production occurs in the bone marrow. There, the process is shielded from the ultraviolet light or starker temperature changes experienced out of the water.

It is possible that this difference evolved long before the first animal with a backbone crawled out of the aquatic environment and faced new, harsher conditions: yet very little fossil evidence exists to support this idea. A definitive answer demands a close examination of fossils from the water-to-land transition including lobe-finned fish and early limbed vertebrates. To support the production of red blood cells, their fin and limb bones would have needed an internal cavity that can house a specific niche that opens onto a complex network of blood vessels.

To investigate this question, Estefa et al. harnessed the powerful x-ray beam produced by the European Synchrotron Radiation Facility and imaged the fin and limb bones from fossil lobe-finned fish and early limbed vertebrates. The resulting three-dimensional structures revealed spongy long bones with closed internal cavities where the bone marrow cells were probably entrapped. These could not have housed the blood vessels needed to create an environment that produces red blood cells.

In fact, the earliest four-legged land animals Estefa et al. found with an open marrow cavity lived 60 million years after vertebrates had first emerged from the aquatic environment, suggesting that blood cells only began to be created in bone marrow after the water-to-land transition. Future work could help to pinpoint exactly when the change in blood cell production occurred, helping researchers to identify the environmental and biological factors that drove this change.

early evolution of the bone marrow and long-bone growth in fossils to contextualise these hypotheses.

Tetrapod long bones are regionalised in three parts mirrored from midshaft (*Figure 1*): (1) the middle of the shaft is called *diaphysis*; (2) the *metaphyses* are located at each extremity of the shaft

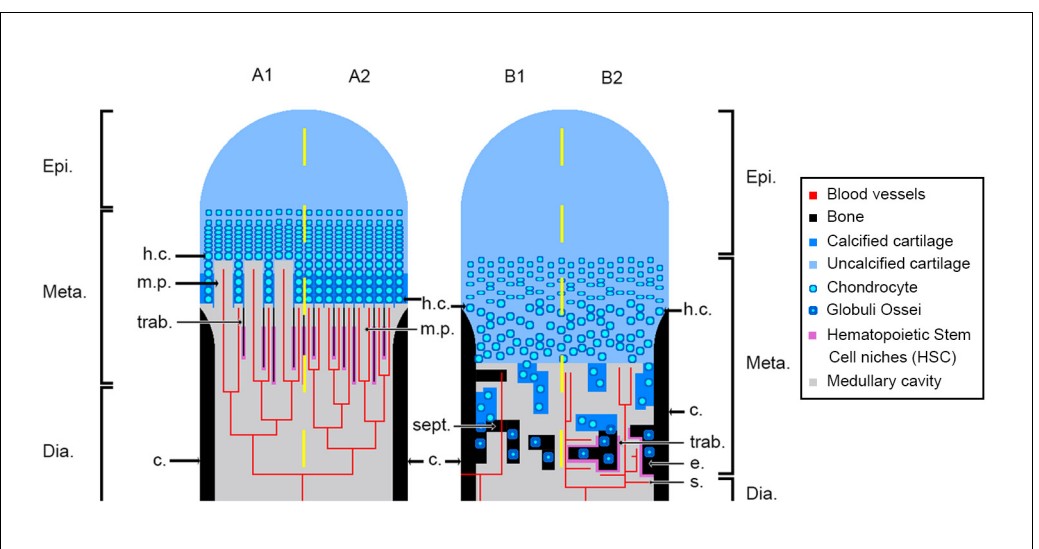

**Figure 1.** Schematic drawing of the long-bone epiphyses of extant amniotes (**A**) and amphibians (**B**). Four conditions are figured here. They are separated by yellow dashed lines: A1, condition in crocodiles (interpreted from **Haines, 1938**); A2, condition in mammals at an early developmental stage before the appearance of the secondary ossification centre (**Anderson and Shapiro, 2010**; **Tanaka, 1976**); B1, condition in *Triturus* (*Cynops*) *pyrrhogaster* (**Quilhac et al., 2014**; **Tanaka, 1976**); B2, condition in *Rana catesbeiana* (**Francillon, 1981**; **Tanaka, 1976**). Abbreviations: c., cortex; Dia., diaphysis; e., endosteal bone; Epi., epiphysis; h.c., hypertrophied chondrocytes; Meta., metaphysis; m.p., marrow process; s., sinusoids; sept., septum; trab., trabeculae.

and (3) the *epiphyses* start above the ossification notch (i.e. where the cortical bone stops forming), extend beyond the metaphyses, and comprise one or more condyles in the case of concave articulations (*Francillon-Vieillot et al., 1990*). Long bones elongate from the growth plate, which is located in the metaphysis (*Figure 1*). In this region, the cartilage is progressively substituted with bone: this process is called endochondral ossification (*Francillon-Vieillot et al., 1990*; *Hall, 2005*).

In extant amniotes, long-bone elongation results from the proliferation of longitudinal columns of hypertrophic cartilage cells, called hypertrophic chondrocytes (*Francillon-Vieillot et al., 1990*; *Haines, 1942*; *Xie et al., 2020*; *Figure 1A*). The latter express collagen type X which facilitates the calcification of the surrounding matrix (*Gudmann and Karsdal, 2016*; *Lüllmann-Rauch, 2015*). To do so, the hypertrophic chondrocytes secrete matrix vesicles containing calcium phosphate crystals (*Amizuka, 2012*; *Anderson and Shapiro, 2010*). The vesicles align longitudinally along the septa. The crystals penetrate the vesicle membranes to form stellate clusters of needle-shaped apatite in the extra cellular matrix (*Amizuka, 2012*). The mineralisation thus propagates following the longitudinal organisation of the septa (*Amizuka, 2012*; *Anderson and Shapiro, 2010*; *Figure 1A*). Blood vessels and marrow processes invade the growth plate along these columns of hypertrophic cartilage (*Lüllmann-Rauch, 2015*; *Figure 1A*). Lytic enzymes secreted by the bone-marrow cells degrade the cartilage matrix, which is progressively substituted by bone deposition (*Lüllmann-Rauch, 2015*; *Suzuki et al., 1981*). Growth factors, such as the vascular endothelial growth factor (VEGF), trigger cartilage calcification and regulate endochondral ossification through stimulation of blood-vessel ingrowth into the diaphysis (*Gerber et al., 1999*). The lines of calcifying stellate clusters of crystals therefore form a scaffold for future trabecular bone deposition (*Amizuka, 2012*). This results in the formation of a bony mesh of longitudinal trabeculae (*Figure 1A*), which is progressively incorporated into the metaphysis where haematopoietic stem cell (HSC) niches (*Figure 1A*) are located (*Calvi et al., 2003*; *Zhang et al., 2003*) in the close vicinity of trabecular/endothelial surfaces (*Gong, 1978*; *Nilsson et al., 2001*; *Wilson and Trumpp, 2006*). HSC form localised niches whose environment is greatly controlled and regulated (*Orkin and Zon, 2008*; *Sipkins et al., 2005*; *Zhang et al., 2003*). Often in mature animals the growth plate disappears, causing the senescence of long-bone elongation (*Kilborn et al., 2002*). In most amniotes, the trabecular mesh in the metaphysis can be vastly remodelled (*Haines, 1975*). HSC can thereafter be observed adjacent to epiphyseal trabeculae (*Askenasy and Farkas, 2002*).

In extant urodeles (e.g. *Pleurodeles waltl*, *De Ricqlès, 1964*; *De Ricqlès, 1965*), the elongation of limb bones differs from the process in amniotes (*Francillon-Vieillot et al., 1990*; *Haines, 1938*; *Figure 1B*). Unlike mammals, endochondral ossification starts at a later stage in urodeles (*De Ricqlès, 1964*). The diaphyseal cartilaginous matrix is first hollowed by the formation of lacunae that are subsequently filled in with bone marrow, far before endochondral ossification starts (*De Ricqlès, 1964*). In mammals and birds (e.g. mouse, *Zelzer et al., 2002*; chicken, *Carlevaro et al., 2000*), VEGF initiates vascular ingrowth before endochondral ossification starts. In the amphibian *Bufo gargarizan*, a peak of VEGF expression is present in the hindlimb at metamorphic climax (*Gao et al., 2018*) paralleling an increase of endochondral ossification activity (*Bo et al., 2018*). VEGF would therefore seem to play a major role in amphibian long-bone endochondral ossification as well, but this role still needs to be characterised. The growth plate in the metaphysis of urodeles exhibits no aligned columns of hypertrophic cartilage cells or drastically reduced alignment of a few cells at most (*De Ricqlès, 1965*; *Dickson, 1982*; *Felisbino and Carvalho, 1999*; *Felisbino and Carvalho, 2001*; *Figure 1B*). Contrary to amniotes, when present, these aligned columns of hypertrophic cartilage do not constitute the location where the ossification takes place (*Figure 1B*). Instead, the mineralisation front is located in the underlying areas of the growth plate (i.e. in a layer of non-oriented hypertrophic cartilage or stratified non-oriented hypertrophic cartilage, *De Ricqlès, 1965*). There, after erosion of the cartilage, mineralisation occurs in urodeles via the formation of globular structures called *globuli ossei* (*Figure 1B*; *De Ricqlès, 1965*; *Quilhac et al., 2014*) and spherical mineralisation around them (forming Liesegang's rings, *Francillon-Vieillot et al., 1990*). *Globuli ossei* are either (1) opened hypertrophic cartilaginous cells which died and were subsequently invaded by a cell from the blood/marrow system to initiate mineralisation or (2) uneroded hypertrophic cartilaginous cells modified into active cells which synthesise bone-like collagen fibrils (of intermediate size between type II of the cartilage and type I of the bone, *Quilhac et al., 2014*). The endochondral ossification therefore does not produce a longitudinally-oriented trabecular network (*Figure 1B*), but forms instead a light reticular mesh (*De Ricqlès,*

*1965*; *Quilhac et al., 2014*; *Sanchez et al., 2008*), rich in *globuli ossei* (*De Ricqlès, 1964*; *Haines, 1938*; *Quilhac et al., 2014*; *Sanchez et al., 2010a*). The epiphyses of urodeles remain cartilaginous while they often ossify in anurans (*Castanet et al., 2003*; *Francillon, 1981*; *Sanchez et al., 2008*). This is probably an adaptation to a demanding jumping locomotion and/or heterochronic mechanisms relevant to this clade (*Francillon, 1981*). In the medullary cavity of their long bones however, the cartilaginous cells hypertrophy with no preferential orientation as in urodeles (*Dickson, 1982*; *Felisbino and Carvalho, 1999*; *Felisbino and Carvalho, 2001*; *Miura et al., 2008*; *Rozenblut and Ogielska, 2005*). The resulting spongiosa is largely reduced (even quite often absent, *Francillon, 1981*). The epiphyseal cartilage hangs over the ossification notch and the shaft (*Francillon, 1981*) to ossify straight after the metamorphosis (*Miura et al., 2008*; *Rozenblut and Ogielska, 2005*). The function of bone marrow in amphibian long bones also differs from the function in extant amniotes. Indeed, in amphibians, the sites for haematopoiesis almost exclusively comprise the thymus, spleen and liver (*Akiyoshi and Inoue, 2012*; *Hightower and Pierre, 1971*). Bone marrow only plays a role of haematopoiesis in a few amphibian species (e.g. *Xenopus laevis*, *Rana catesbeiana*, *Tanaka, 1976*; *Phillobates terribilis*, *Dendrobates tinctorius*, *Kapp et al., 2018*). In these cases, haematopoiesis occurs in endosteal regions of the diaphysis between sinusoids (i.e. fenestrated capillaries; *Figure 1B2*) and endosteum (*Tanaka, 1976*). No HSC has been observed so far in the epiphysis of frogs.

Is the urodele model the plesiomorphic or the derived condition for tetrapod long-bone elongation and bone-marrow function? Very little attention has been given to these aspects of limb-bone evolution. On the one hand, some authors suggest that the amniote-like elongation process may have been the primitive state (*Haines, 1942*) but no fossil evidence was provided. On the other hand, early tetrapods had cartilaginous epiphyses like extant urodeles. Could that be an indication for a urodele-like primitive condition (e.g. *Sanchez et al., 2008*; *Sanchez et al., 2010a*)? This debate relied on the absence of evidence from stem-tetrapod data. Recently, palaeohistological studies revealed a fan-like longitudinal trabecular arrangement in the long-bone metaphysis of the 380-million-year-old lobe-finned fish *Eusthenopteron* (*Sanchez et al., 2014*), and in the 365-million-year-old limbed stem-tetrapod *Acanthostega* (*Sanchez et al., 2016*). These patterns would result from the same elongation process as in amniotes and would rather suggest that amphibians exhibit a derived condition. Using three-dimensional (3D) virtual histology based on propagation phase-contrast X-ray synchrotron radiation micro-computed tomography (PPC-SRμCT), as well as classical thin-section histology, we herein investigate several stem amphibians and stem amniotes to provide the first glimpses for characterising the early evolution of long-bone elongation and bone-marrow roles.

## Results

The diaphyseal and metaphyseal microarchitectures of the stem tetrapods *Eusthenopteron* and *Hyneria* were described by *Sanchez et al., 2014* and *Kamska et al., 2018*, respectively. The diaphysis of *Discosauriscus* (*Sanchez et al., 2008*), *Apateon* (*Sanchez et al., 2010a*; *Sanchez et al., 2010b*), *Metoposaurus* (*Konietzko-Meier and Sander, 2013*) and *Seymouria* (*Estefa et al., 2020*) were thoroughly described, but the metaphyseal organisation of their humeri was only succinctly mentioned in the cited articles. Here, we provide a detailed description of them (*Table 1*) in 3D when possible.

### *Apateon caducus*, juvenile specimen GPIM-N 1297, humerus

As the humerus was crushed (*Figure 2A*), only a small region of the metaphysis could be sectioned and visualised (*Figure 2B*). Nevertheless, a relatively complete sequence of calcification (extending over 600 μm) can be described here. The upper part of the section reflects the irregular surface of the calcification front (separating the unpreserved eroded non-calcified cartilage from the preserved calcified cartilage) (*Figure 2B*). Under this region, obvious figures of *globuli ossei* are entrapped in Liesegang's rings (g.o. and l.r., *Figure 2Bb1-2*). They are numerous and unevenly arranged. Their sizes (ranging from 9 to 15 μm in diameter) seem as well unevenly distributed. The trabeculae are very few in this thin section (t., *Figure 2Bb2*).

**Table 1.** Table summarising the material used.
Skull length measurements and ontogenetic stages determined by *Berman et al., 1987b*; *Sanchez et al., 2008*; *Sanchez et al., 2010a* and *Klembara et al., 2006*.

| Species | Collection number | Skull length (cm) | Ontogenetic stage | Bone |
|---|---|---|---|---|
| *Apateon caducus* | GPIM-N 1297 | 1.52 | Juvenile | Humerus |
| | | | | Radius |
| | | | | Ulna |
| | GPIM-N 1572 | Estimated to 1.60 | Adult | Radius |
| | | | | Ulna |
| *Apateon pedestris* | SMNS 54981 | 0.86 | Adult | Humerus |
| | | | | Radius |
| | | | | Ulna |
| | SMNS 54988 | 1.06 | Adult | Humerus |
| | | | | Radius |
| | | | | Ulna |
| *Seymouria sanjuanensis* | MNG 7747 | 5.6 | Juvenile | Humerus |
| | CM 28597 | 8.8 | Adult | Humerus |
| *Discosauriscus austriacus* | SNM Z 15568 | 6.2 | Subadult | Humerus |
| *Metoposaurus* sp. | MUZ PGI OS-220/171 | - | Subadult or adult | Humerus |

### A. *caducus*, juvenile specimen GPIM-N 1297, radius and ulna

Both bones exhibit large sequences of cartilage calcification which spread over more than a third of the total bone length on each side of the long bone (*Figure 2C*). The mineralisation front (m.f., *Figure 2C*) is located relatively far under the ossification notch (400 µm) (o.n., *Figure 2C*). Numerous *globuli ossei* can be visualised in the metaphysis (g.o., *Figure 2Cc1,3*). They are unevenly distributed and their size ranges between 8 and 25 µm. Clusters of chondrocytes can be observed (c.c., *Figure 2Cc3*). The top of the epiphysis probably exhibited a uniform matrix of uncalcified cartilage before the fossilisation that was not preserved afterwards. The mesh of ossified trabeculae is very scattered and shows no preferential orientation (t., *Figure 2Cc1*).

### A. *caducus*, adult specimen GPIM-N 1572, radius and ulna

The epiphysis and metaphysis of the radius and ulna of this individual (*Figure 3A*) are more hollowed than those of the specimen GPIM-N 1297, with less cartilaginous matrix between the mineralised trabeculae (*Figure 3B*). Fewer *globuli ossei* are visible (g.o., *Figure 3Bb1*). Instead, large empty lacunae can be observed (75 µm) (e.l., *Figure 3Bb1*). Large bays of erosion open as well between these lacunae (e.b., *Figure 3Bb2*). The process of mineralisation therefore seems more advanced but no obvious trabecular organisation can be observed.

### *Apateon pedestris*, adult specimen SMNS 54981, humerus

In the humerus of SMNS 54981 (*Figure 4A*), the process of mineralisation seems relatively advanced as the *globuli ossei* only remain along a few mineralised trabeculae (g.o., *Figure 4B*). They are 16 µm large. In the metaphysis, the cartilage has been removed by erosional process (e.b., *Figure 4B*). The uncalcified cartilage in the epiphysis has not been preserved during the fossilisation (at least in this slide).

### A. *pedestris*, adult specimen SMNS 54981, radius and ulna

The quantity of calcified cartilage is higher in the zeugopod (i.e. radius and ulna) than in the stylopod (i.e. humerus) (*Figure 4Cc1-2*). Most of the uncalcified cartilage has been eroded. The calcified cartilage is hollowed, thereby forming multiple bays of erosion (e.b., *Figure 4Cc2*). Nevertheless, the

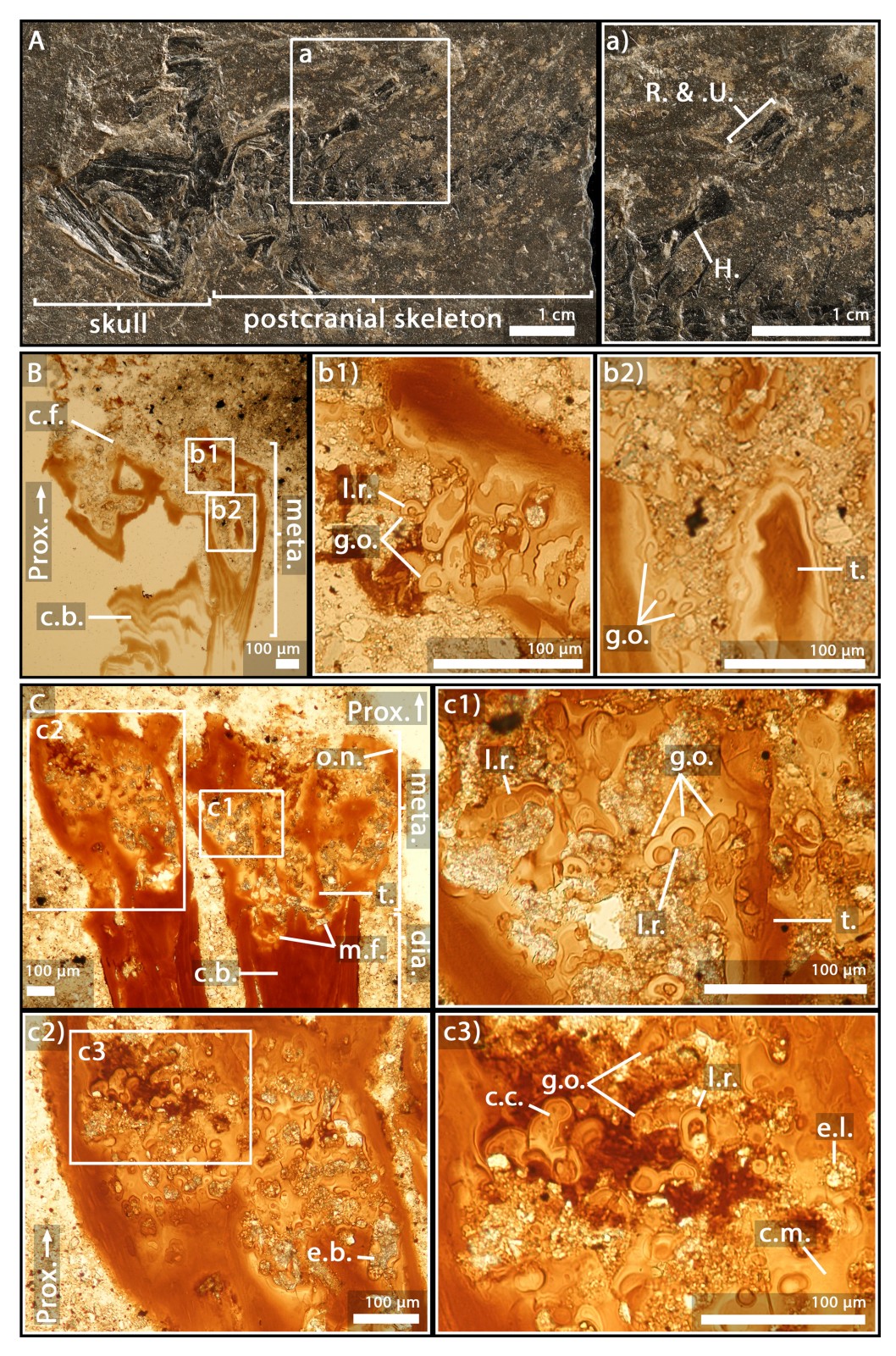

**Figure 2.** Juvenile specimen of *Apateon caducus*, GPIM-N 1297. (**A**) Skeleton. (**a**) Right limb. (**B**) Epiphyseal and metaphyseal histology of the proximal end of the humerus. (**C**) Epiphyseal and metaphyseal histology of the proximal end of the radius (**c2-3**) and ulna (**c1**). Abbreviations: c.b., cortical bone; c.c., cluster of chondrocytes; c.f., calcification front; c.m., cartilage matrix; dia., diaphysis; e.b., erosion bay; e.l., erosion lacunae; g.o., globuli ossei; H.,

*Figure 2 continued on next page*

*globuli ossei* remain connected to each other by calcified-cartilage trabeculae (c.c.t., *Figure 4Cc2*) or mineralised trabeculae (m.t., *Figure 4C*) present in the metaphysis.

### *A. pedestris*, adult specimen SMNS 54988, humerus

This thin section in the humerus of SMNS 54988 (*Figure 5A*) shows a very remodelled bone with large bays of erosion in the cartilaginous matrix (e.b., *Figure 5Bb*) and only a few remaining *globuli ossei* at the surface of the bone trabeculae (g.o., *Figure 5Bb*). Most of the cartilaginous matrix has been eroded. There is no preserved cartilage in the epiphysis. The bony trabeculae have no preferential orientation.

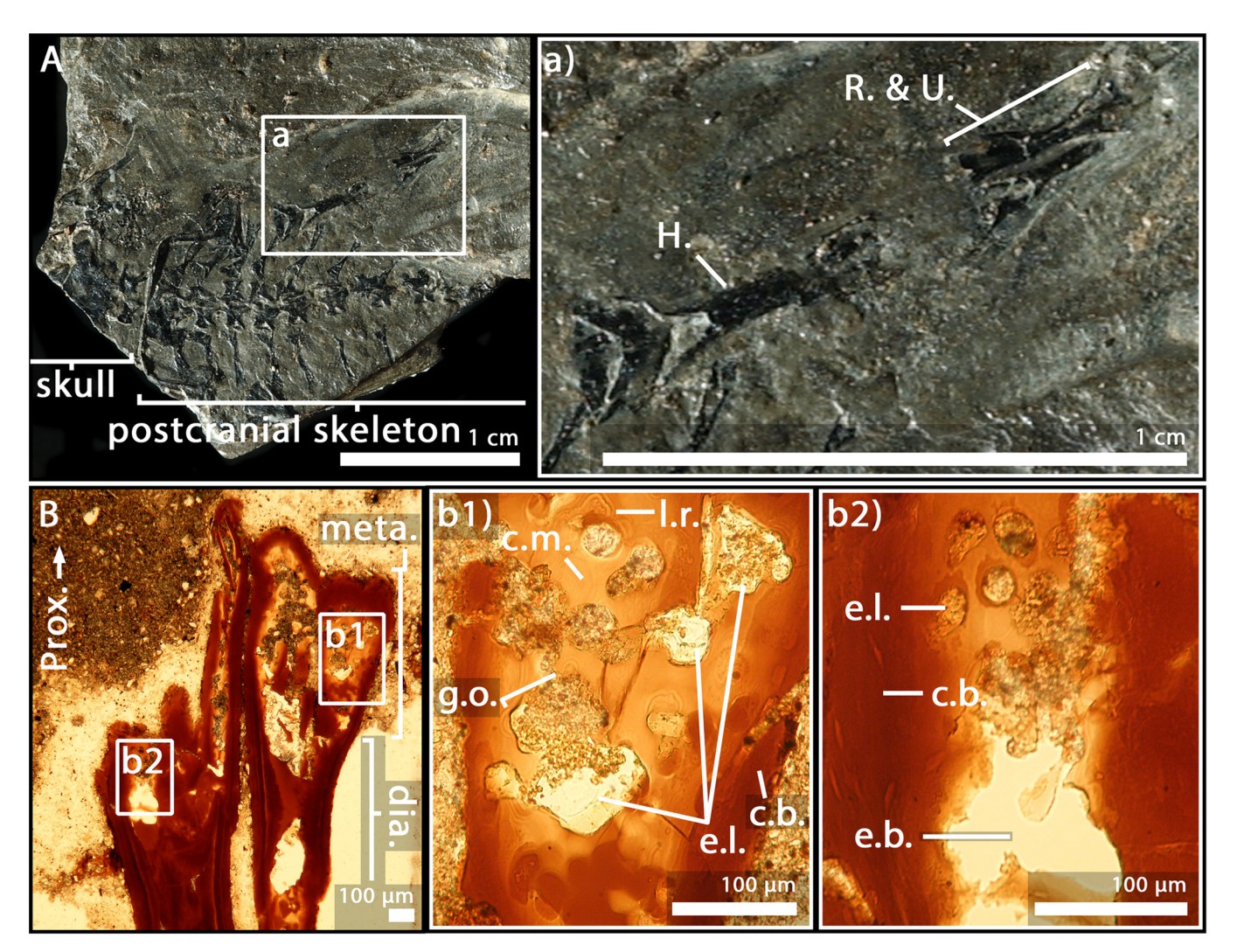

**Figure 3.** Adult specimen of *Apateon caducus*, GPIM-N 1572. (A) Skeleton. (a) Right limb. (B) Epiphyseal and metaphyseal histology of the proximal end of the radius (b1) and ulna (b2). Abbreviations: c.b., cortical bone; c.m., cartilage matrix; dia., diaphysis; e.b., erosion bay; e.l., erosion lacunae; g.o., globuli ossei; H., humerus; l.r., Liesegang's rings; meta., metaphysis; Prox., proximal end; R. and U., radius and ulna.

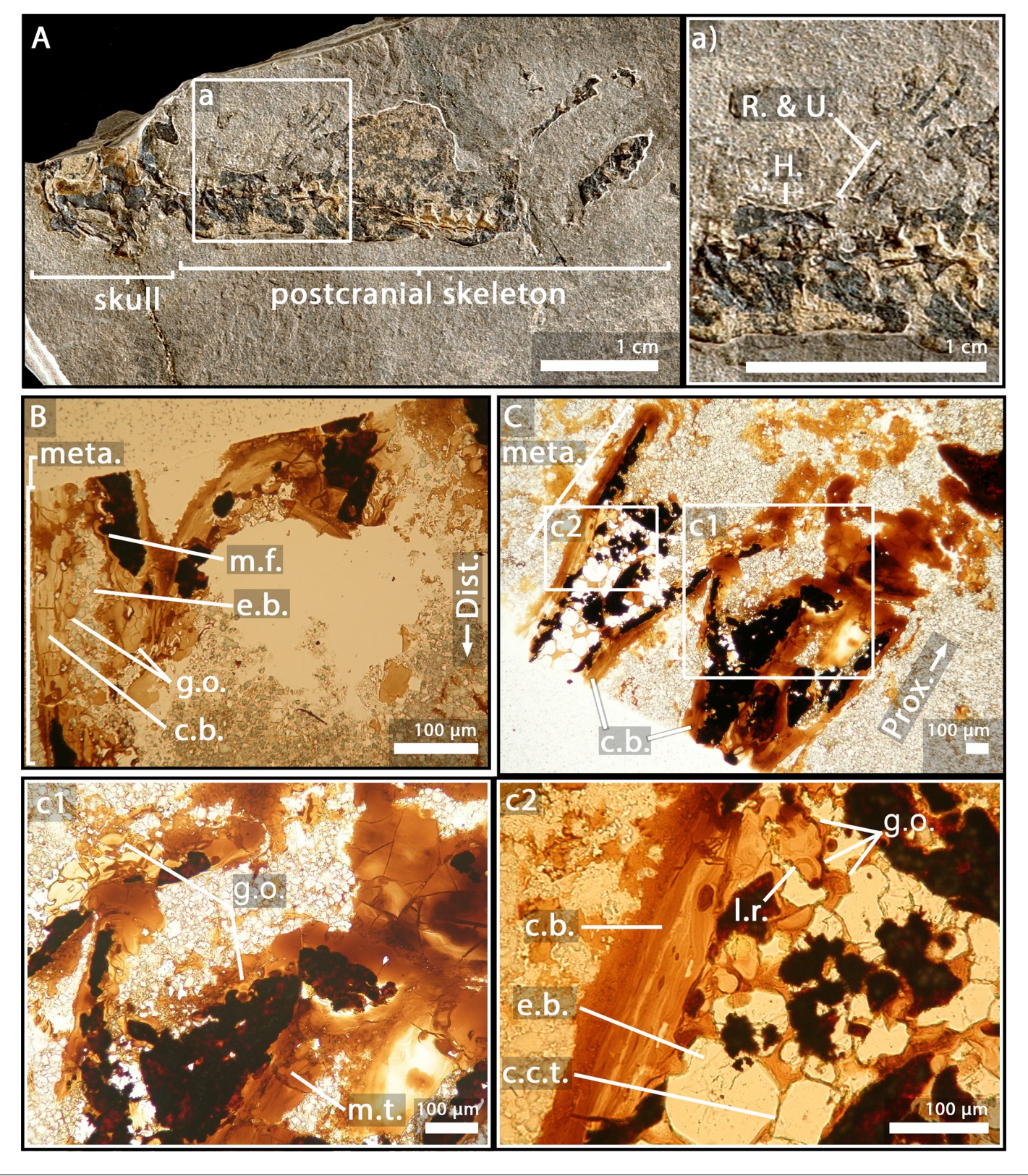

**Figure 4.** Adult specimen of *Apateon pedestris*, SMNS 54981. (**A**) Skeleton. (**a**) Right limb. (**B**) Epiphyseal and metaphyseal histology of the distal end of the humerus. (**C**) Epiphyseal and metaphyseal histology of the proximal end of the radius (**c2**) and ulna (**c1**). Abbreviations: c.b., cortical bone; c.c.t.,

*Figure 4 continued on next page*

*Figure 4 continued*

calcified-cartilage trabecula; Dist., distal end; e.b., erosion bay; g.o., globuli ossei; H., humerus; l.r., Liesegang's rings; meta., metaphysis; m.f., mineralisation front; m.t., mineralised trabecula; Prox., proximal end; R. and U., radius and ulna.

### *A. pedestris*, adult specimen SMNS 54988, radius and ulna

As for the zeugopod of the specimen SMNS 54988 (*Figure 5Aa*), the *globuli ossei* seem to be replaced by large empty lacunae (30 μm, e.l., *Figure 5C*). A certain amount of uncalcified cartilage has been eroded in the distal epiphyses and metaphyses. Nevertheless, a large amount of cartilage is still present in the proximal metaphyses of both long bones (c.m., *Figure 5C*). No or very few trabeculae can be observed.

### *Metoposaurus* sp., (sub-)adult specimen MUZ PGI OS-220/171, humerus

Transverse thin sections were made in the metaphysis of the femur (*Konietzko-Meier and Sander, 2013*) of *Metoposaurus diagnosticus krasiejowensis* (*Sulej, 2002*) recently re-diagnosed as

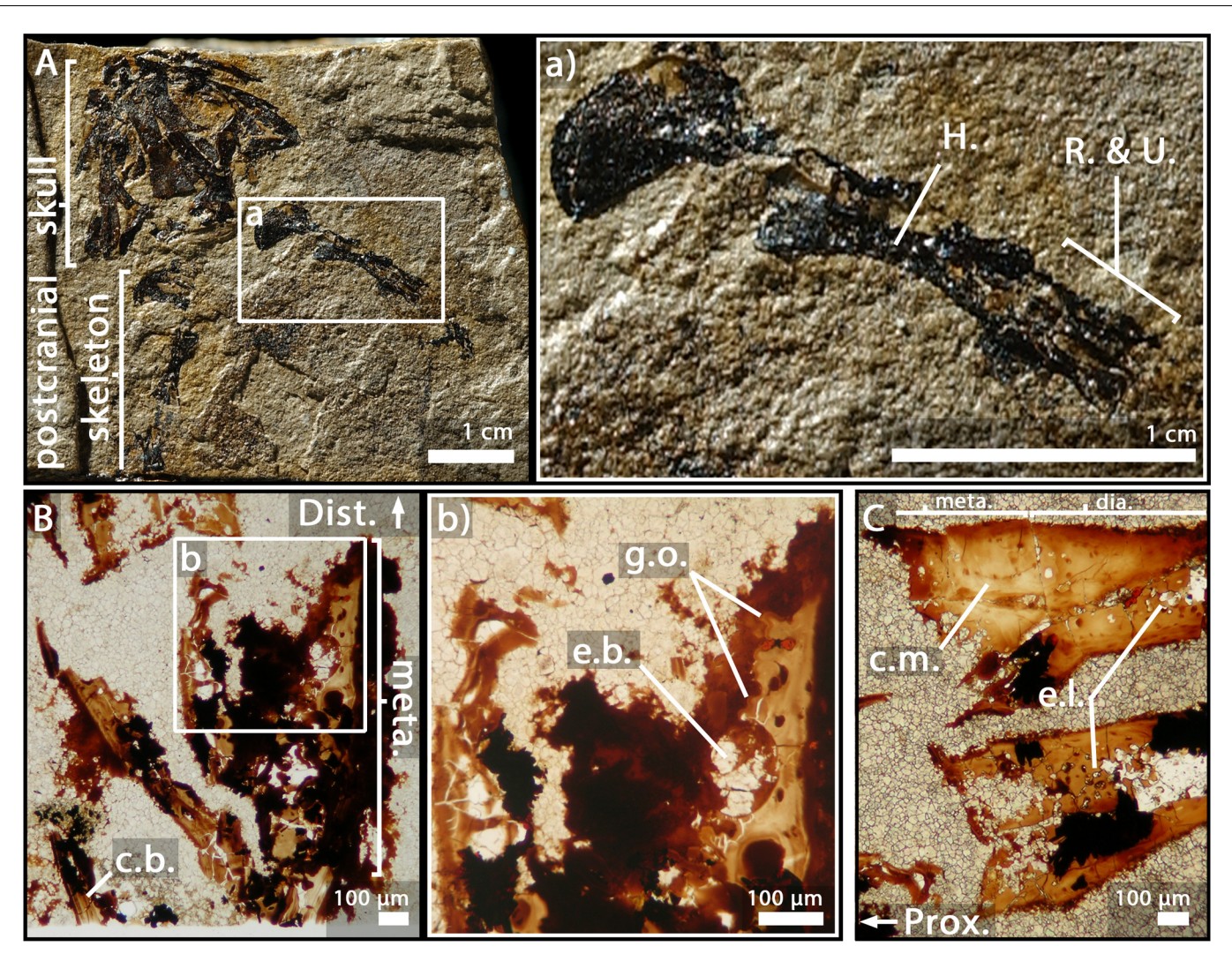

**Figure 5.** Adult specimen of *Apateon pedestris*, SMNS 54988. (A) Skeleton. (a) Right limb. (B) Epiphyseal and metaphyseal histology of the distal end of the humerus. (C) Epiphyseal and metaphyseal histology of the proximal end of the radius and ulna. Abbreviations: c.b., cortical bone; c.m., cartilage matrix; dia., diaphysis; Dist., distal end; e.b., erosion bay; e.l., erosion lacunae; g.o., globuli ossei; H., humerus; meta., metaphysis; Prox., proximal end; R. and U., radius and ulna.

*Metoposaurus krasiejowensis* (*Brusatte et al., 2015*). They revealed a dense trabecular mesh. The longitudinal virtual thin sections, made with PPC-SRµCT and presented here, were made in the proximal and distal metaphyses of a humerus of *Metoposaurus* sp. and confirm the presence of a dense trabecular mesh in the overall humerus (*Figure 6Aa1-2*). Additionally, a directional coloured light effect (cf. Materials and method section, *Sanchez et al., 2014*) shows that this mesh is oriented longitudinally and exhibits a fan-like shape in the metaphyses (purple trabeculae, *Figure 6Aa2*). The trabecular mesh covers the entire volume of the metaphysis and spreads into the diaphysis (*Figure 6Aa1-2*). The mineralisation front (m.f., *Figure 6Aa1*) contacts the sediment in which the bone is embedded (*Figure 6Ba*). The surface of the mineralisation front is irregular. No ossified epiphysis was found, thereby suggesting that a cartilaginous cap was probably covering the bone. This cap did not preserve over the fossilisation. In the metaphysis, the trabeculae are homogeneously distributed (t., *Figure 6Bb*). Some remnants of calcified cartilage are visible through Liesegang's rings forming within the cartilage remaining between the metaphyseal trabeculae (*Figure 6Bc*). The mean thickness of the trabeculae is 117 µm (*Table 2*). Tubular structures can be observed (m.p., *Figure 6Ba*). They end blindly at the location of the mineralisation front. They are well defined tubes (248 µm in diameter, *Table 2*), although anastomosed. They ossified through endochondral ossification. These tubes are locally slightly eroded (*Figures 6Ba* and *10A*). The size of these tubes, their intimate connection to each other and their location strongly support their identification as marrow processes (*Haines, 1938*).

### *Seymouria sanjuanensis*, juvenile specimen MNG 7747, humerus

This specimen was investigated using PPC-SRµCT. The spongiosa occupies the entire bone area (*Figure 7A*). The metaphyseal trabeculae are about four times thinner (25 µm on average) than the diaphyseal trabeculae (94 µm on average, *Table 2* and *Estefa et al., 2020*). A longitudinal section reveals that the trabecular mesh becomes denser towards the distal and proximal ends of the bone (*Figure 7Aa1-2*). As the shape of the bone widens and flattens from midshaft towards the metaphyseal surfaces, the longitudinal trabeculae tilt, thereby forming a fan-like configuration (*Figure 7Aa2*). In the metaphysis, the trabecular mesh is mostly arranged longitudinally (obviously appearing purple and progressively shifting to green as the deltopectoral crest tilts to 90 degrees, *Figure 7Aa2*) although a few anastomoses (highlighted in green in most of the metaphysis apart from the tilted region of the deltopectoral crest, *Figure 7Aa2*) run radially. A few remnants of calcified cartilage (*Francillon-Vieillot et al., 1990*) are very rarely visible (*Estefa et al., 2020*). Marrow processes form an intricate network, while anastomosing to each other, and connecting to cavities of irregular shapes and sizes (*Figure 7Ba,c* and *10B*). These tubular structures are around 100 µm in diameter under the mineralisation front (*Table 2*). They contact each other when they reach the mineralisation front (*Figure 10B*). No ossified epiphysis was found. The humeral epiphysis was probably not preserved due to being unmineralised cartilage.

### *S. sanjuanensis*, adult specimen CM 28597, humerus

A longitudinal virtual thin section from the PPC-SRµCT data shows that the trabecular network remains relatively dense in the metaphyses (*Figure 8A*) at the adult stage. The trabecular mesh is longitudinally and radially oriented like a fan although slightly less organised than in the juvenile specimen (purple trabeculae, *Figure 8Aa2*). The trabeculae appear to be more remodelled, leaving large cavities resulting from an intense erosional process (*Figure 8Aa2*). The cortex is almost inexistent in the metaphysis (*Figure 8Bb*). The thickness of the trabeculae averages 30 µm (*Table 2*, *Estefa et al., 2020*), which is equivalent to the thickness of the trabeculae in the juvenile metaphysis (MNG 7747). There is no endosteal bone on the surface of the medullary cavity. Very few remnants of calcified cartilage were found in the metaphysis of the adult humerus, that is in much lower frequency than in the juvenile specimen (*Estefa et al., 2020*). The spongiosa contains a few longitudinal interconnected marrow processes (100 µm in diameter, *Figure 8Bb-c* and *10C*, *Table 2*). When present, these processes exhibit the same distribution as in the juvenile humerus. The epiphyses were not ossified. A resting surface (yellow arrow, *Figure 8Aa2*), red arrows, (*Figure 8—figure supplement 1A*), can be observed 2-to-4 mm under the mineralisation front. It is observed as well in the distal metaphysis (*Figure 8—figure supplement 1A*). This resembles Harris lines identified in mammals (*Garn et al., 1968*; *Harris, 1933*) and birds (*Wegner, 1874*). Although these lines are very

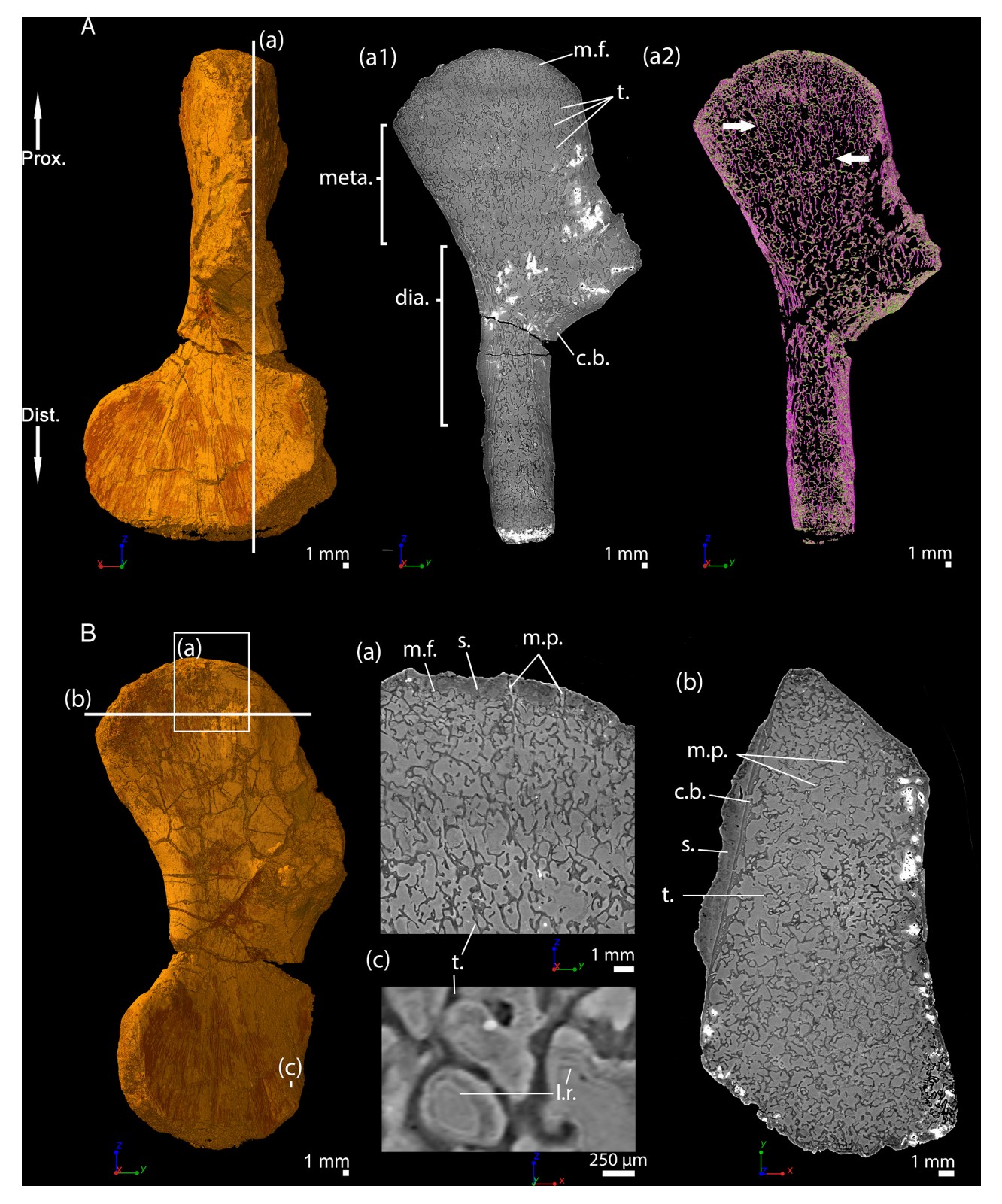

**Figure 6.** Left humerus of a (sub-)adult specimen of *Metoposaurus* sp., MUZ PGI OS-220/171 imaged using PPC-SRμCT. (**A**) Frontal view. (**a1**) Longitudinal virtual thin section (40 μm thick) and (**a2**) longitudinal virtual thin section of the segmented model of the bone (50 μm thick). The longitudinally-oriented trabeculae are highlighted in purple (white arrows), while the transversally-oriented trabeculae appear in green. (**B**) Ventral view. (**a**) Longitudinal virtual thin section of the proximal metaphysis (40 μm thick), (**b**) transverse virtual thin section made in the metaphysis and (**c**)
*Figure 6 continued on next page*

*Figure 6 continued*

longitudinal thin section made in the distal metaphysis. Abbreviations: c.b., cortical bone; dia., diaphysis; Dist., distal end; l.r., Liesegang's rings; meta., metaphysis; m.f., mineralisation front; m.p., marrow process; Prox., proximal end; s., sediment; t., trabeculae.

common in mammals (including extant and extinct taxa, *Duckler and Van Valkenburgh, 1998*), they have not been comprehensively studied in other groups. We find that they can also be encountered in groups with no secondary ossification centre such as chelonians (e.g. *Centrochelys sulcata*, *Figure 8—figure supplement 1B*) and crocodilians (e.g. *Crocodylus niloticus*, P.T. pers. obs.). Harris lines seem to result from both short- and long-term pressures (e.g. starvation – *Park, 1964*; disease and deficiencies – *Duckler and Van Valkenburgh, 1998*).

### *Discosauriscus austriacus*, subadult specimen SNM Z 15568, humerus

Thin sections were made and described in the distal metaphysis of the femur of the specimen KO224 of *D. austriacus* (*Sanchez et al., 2008*). They revealed a dense trabecular mesh and the absence of calcified cartilage. Virtual thin sections from three-dimensional PPC-SRμCT scans in the humerus SNM Z 15568 complete these observations despite the fact that the metaphyseal spongiosa is partly crushed (*Figure 9Aa2*). As the femur KO224 (*Sanchez et al., 2008*), the humerus SNM Z 15568 exhibits a dense trabecular mesh in both the proximal and distal metaphyses with a fan-like trabecular orientation (purple trabeculae, *Figure 9Aa2*). The trabeculae are homogeneously distributed in the metaphysis. They are 54 μm thick on average (*Table 2*). They are eroded at the base of the metaphysis (*Figure 9Aa1-2*). Some tubular marrow processes (average diameter: 111 μm, *Table 2*) open up directly towards the diaphysis into the metaphyseal space of the medullary cavity left vacant after erosion (*Figures 9B* and *10D*). The surface of the mineralisation front is irregular (*Figure 9Ba*). The latter probably was covered by uncalcified cartilage.

## Discussion

The trabecular bone tissues observed in these long bones exhibit characteristics of endochondral ossification (remnants of calcified cartilage, *globuli ossei* and/or columnar trabecular mesh) as seen in stem- (*Sanchez et al., 2014*; *Sanchez et al., 2016*) and crown-tetrapods (*Estefa et al., 2020*; *Francillon-Vieillot et al., 1990*; *Sanchez et al., 2008*; *Sanchez et al., 2010a*).

### Early evolution of tetrapod limb-bone elongation

Although all limb bones studied here have cartilaginous epiphyses, their metaphyseal organisation, and the underlying long-bone elongation processes, can greatly differ between taxa.

The long-bone elongation in *Apateon* probably results from the hypertrophying action of scattered cartilaginous cells in the upper part of the metaphysis (*Figure 11*). The fossils revealed a mineralisation front characterised by a large number of *globuli ossei* (*Figures 2–5*), which progressively replaced these hypertrophic chondrocytes, as in urodeles (*De Ricqlès, 1964*; *De Ricqlès, 1965*; *Haines, 1938*; *Quilhac et al., 2014*). As the uncalcified cartilage is not preserved in *Apateon*, it is

**Table 2.** Microanatomical measurements made on the samples using *VGStudio MAX* (version 3.2, Volume Graphics Inc, Germany). The protocol details are provided by *Estefa et al., 2020*.

| Species | Thickness of the trabeculae (μm) | | Diameter of the marrow processes (μm) |
| --- | --- | --- | --- |
| | Diaphysis | Metaphysis | Metaphysis |
| *Metoposaurus* sp. (Subadult or Adult, MUZ PGI OS-220/171) | 131 | 117 | 248 |
| *Seymouria sanjuanensis* (Juvenile, MNG 7747) | 94 | 25 | 100 |
| *S. sanjuanensis* (Adult, CM 28597) | 79 | 30 | 100 |
| *Discosauriscus austriacus* (Subadult, SNM Z 15568) | 80 | 54 | 111 |

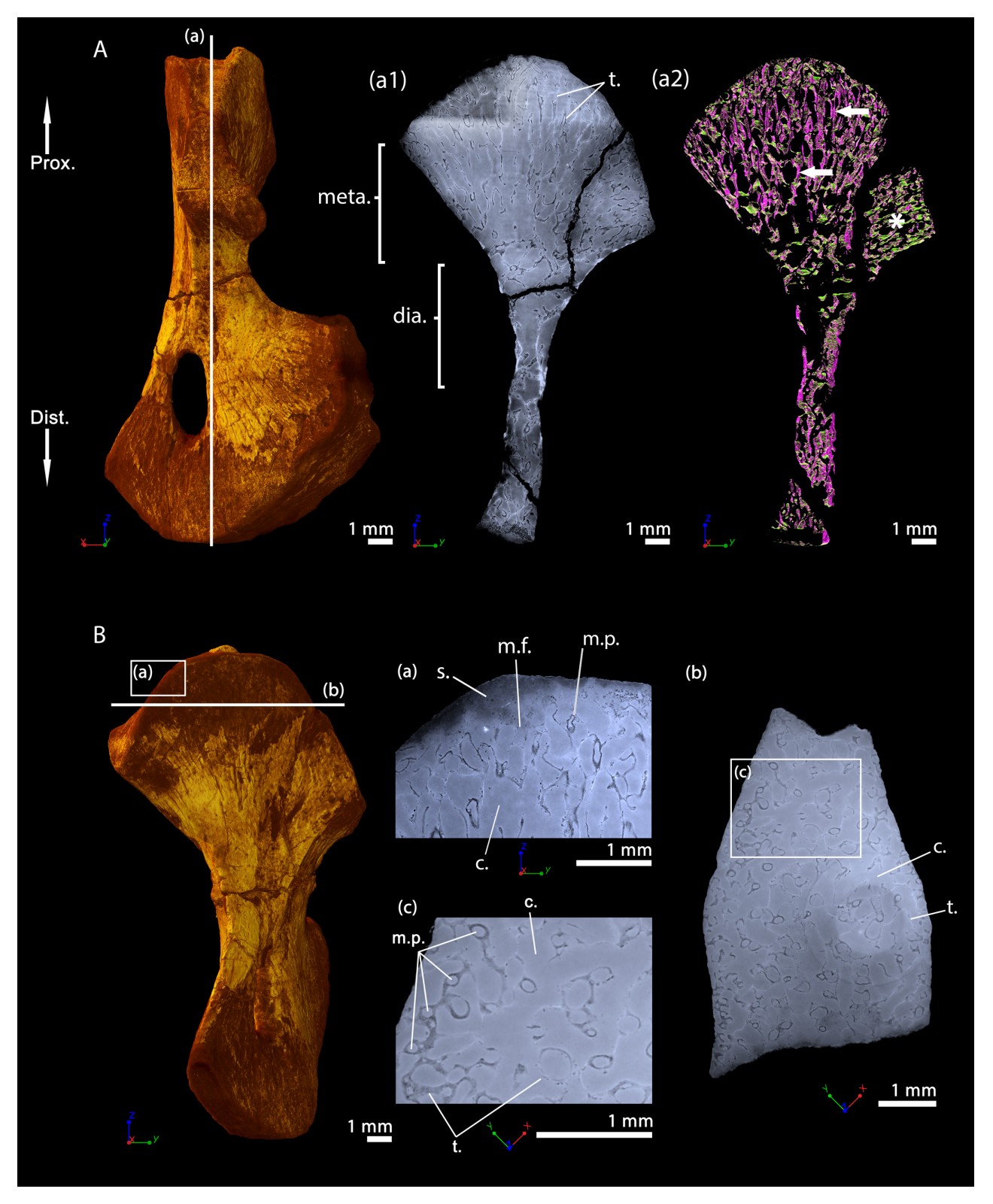

**Figure 7.** Left humerus of a juvenile specimen of *Seymouria sanjuanensis,* MNG 7747 imaged using PPC-SRμCT. (**A**) Frontal view. (**a1**) Longitudinal virtual thin section (40 μm thick), the darker part is an artefact in the original data due to electron reinjection in the synchrotron storage ring (refilling) during the scan and (**a2**) longitudinal virtual thin section of the segmented model of the bone (250 μm thick). The longitudinally-oriented trabeculae (pointed by horizontal arrows) are highlighted in purple, while the transversally-oriented trabeculae appear in green. Note that, due to the shape of the

*Figure 7 continued on next page*

*Figure 7 continued*

metaphysis, the trabeculae exhibit an overall fan-like configuration which progressively tilts to 90 degrees at the location of the deltopectoral crest (Asterisk). For that reason, the longitudinal trabeculae appear green and the transverse trabeculae appear purple at this location. (**B**) Ventral view. (**a**) Longitudinal virtual thin section in the proximal metaphysis (40 µm thick), (**b**) transverse virtual thin section in the metaphysis, the large ring artefact results from the synchrotron electron refilling visible in a1, (**c**) detail of (**b**) showing marrow processes and cavities in transverse section. Abbreviations: c., cavity; dia., diaphysis; Dist., distal end; meta., metaphysis; m.f., mineralisation front; m.p., marrow process; Prox., proximal end; s., sediment; t., trabeculae.

not possible to check whether the growth zone and calcifying zone are distant as in amphibians (*De Ricqlès, 1965*; *Felisbino and Carvalho, 1999*; *Felisbino and Carvalho, 2001*). However, the presence of very few bony trabeculae, with no preferential orientation in the metaphyseal region of *Apateon*'s long bones (*Figures 2–5*) strongly support the idea that the ossification did not occur within a zone of columnar hypertrophic cartilage. Bone elongation was probably followed by mineralisation through *globuli ossei* as previously observed in the temnospondyl dissorophoid *Doleserpeton* (*De Ricqlès, 1963*; *Figure 11*).

On the contrary, the metaphyseal trabecular meshes in the limb bones of *Metoposaurus*, *Seymouria* and *Discosauriscus* all exhibit the same fan-like pattern of longitudinal trabeculae. This implies that the growth plate comprised longitudinal columns of hypertrophic cells where endochondral ossification occurred through columnar cartilage-to-bone substitution (*Francillon-Vieillot et al., 1990*; *Kronenberg, 2003*) as seen in turtles and crocodiles (e.g. *Haines, 1938*; *Haines, 1942*), lepidosaurs (e.g. *Haines, 1969*), dinosaurs (*De Ricqlès, 1968*; *Horner et al., 2000*; *Horner et al., 2001*), birds (e.g. *Horner et al., 2001*) and mammals (e.g. *Jacenko et al., 1993*; *Figure 11*). Although this fan-like trabecular configuration can be greatly remodelled in extant amniotes, it is only slightly remodelled here in the adult *Seymouria* and subadult *Discosauriscus* (*Figures 8* and *9*).

Therefore, our study shows that the amniote-like long-bone elongation is more commonly distributed than previously thought. It is not restrained to the appendicular skeleton of amniotes – including *Discosauriscus* and *Seymouria* (as demonstrated here), *Ophiacodon*, *Dicynodon* and some kannemeyeriids (*De Ricqlès, 1972*; *Haines, 1938*), marine reptiles, *Plesiosaurus* and *Nothosaurus* (*Haines, 1938*) – neither to that of stem tetrapods (*Kamska et al., 2018*; *Sanchez et al., 2014*; *Sanchez et al., 2016*; *Figure 11*). Indeed, our study demonstrates for the first time that certain temnospondyls like *Metoposaurus* also elongated their appendicular skeleton like amniotes (*Figure 11*). We can confidently conclude that (1) endochondral ossification based on the mineralisation of longitudinal columns of hypertrophic cartilage is a primitive process for the elongation of the appendicular skeleton in tetrapods and that (2) ossification through *globuli ossei* is restricted to a limited group of stem- and crown-batrachians (*Figure 11*).

The mineralisation processes in extant amphibians and amniotes largely differ in many points: (1) their timing and microstructural relationships (i.e. ossification dependant on a calcified scaffold in amniotes, *Amizuka, 2012*, but not in amphibians, *Felisbino and Carvalho, 2001*), (2) their initiation (i.e. stacks of hypertrophic cells in amniotes, *Lüllmann-Rauch, 2015*, versus isolated hypertrophic cells in amphibians, *De Ricqlès, 1965*; *Quilhac et al., 2014*) and (3) their molecular mechanisms (i.e. collagen type X secreted in amniotes, *Gudmann and Karsdal, 2016*; versus fibrillar collagens in amphibians, *Quilhac et al., 2014*). How could such distinct mineralisation processes play the same functional role in long-bone elongation in different vertebrates? In addition to a columnar pattern of trabeculae, several remnants of Liesegang's rings could be observed in the fossil limb bones of *Metoposaurus* (*Figure 6Bc* and *Konietzko-Meier and Sander, 2013*) and, to a lesser extent, *Seymouria* (*Estefa et al., 2020*) and stem-tetrapods (i.e. *Hyneria*, *Eusthenopteron* and *Acanthostega*, *Kamska et al., 2018*; *Sanchez et al., 2014*; *Sanchez et al., 2016*). The most parsimonious evolutionary scenario therefore suggests that stem tetrapods were able to produce globular calcified cartilage although they were elongating their bone through a columnar configuration (*Figure 11*). Long-bone elongation exclusively based on the intensive production of *globuli ossei* would have been a derived feature emerging within temnospondyls and restricted to extant batrachians and their close dissorophoid temnospondyl relatives (e.g. *Apateon* and *Doleserpeton*). Within amniotes – including stem amniotes – the globular calcification of the cartilage would have drastically reduced to be fully abandoned to the benefit of an exclusive columnar elongation (*Figure 11*). Two exceptions persist in amniotes: (1) large aquatic animals usually produce a large amount of

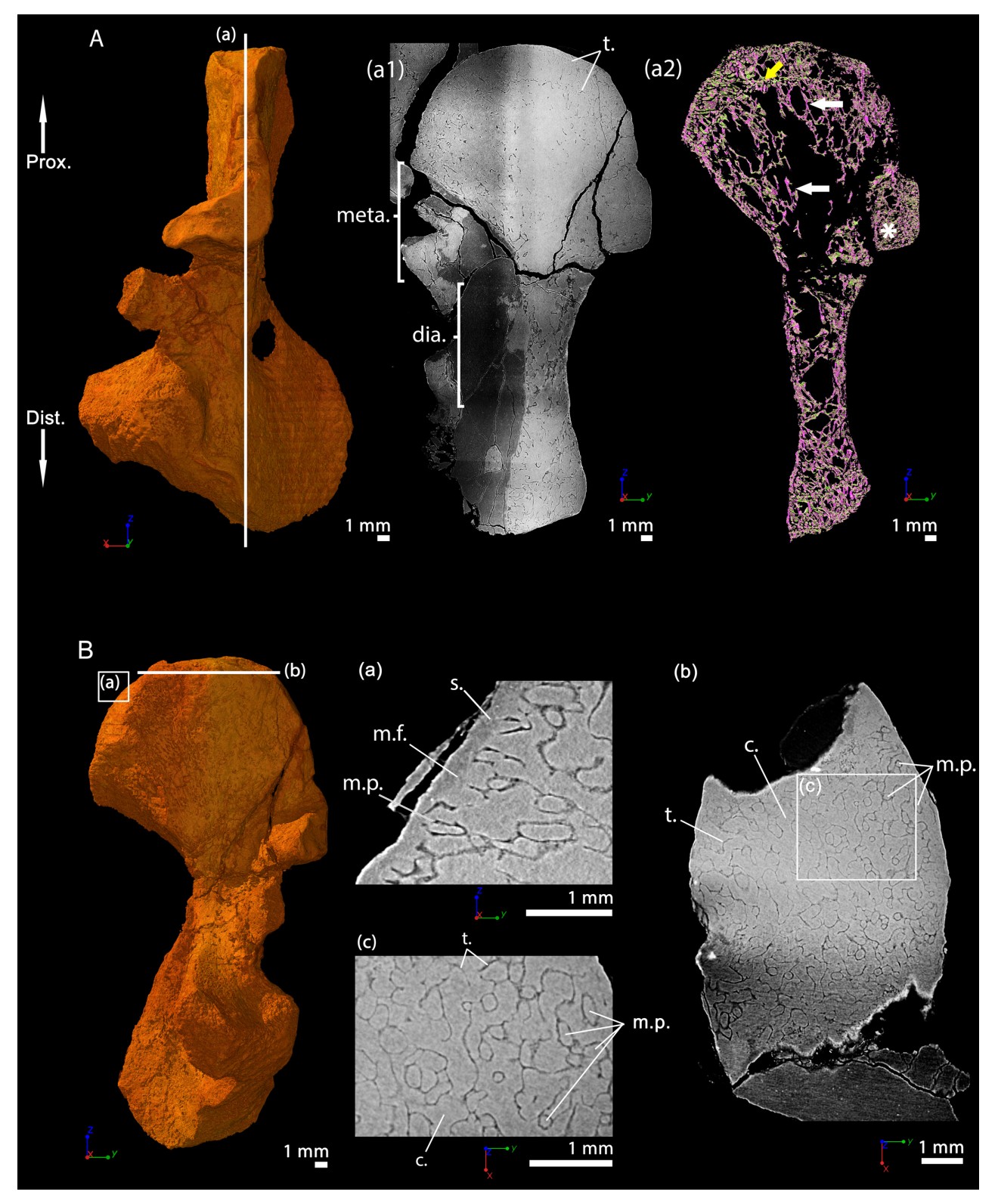

**Figure 8.** Right humerus of an adult *Seymouria sanjuanensis*, CM 28597 imaged using PPC-SRµCT. (**A**) Frontal view. (**a1**) Longitudinal virtual thin section (40 µm thick) and (**a2**) longitudinal section of the segmented model of the bone (450 µm thick). The longitudinally-oriented trabeculae (pointed by horizontal arrows) are highlighted in purple, while the transversally-oriented trabeculae appear in green. Note that, due to the shape of the metaphysis, the trabeculae exhibit an overall fan-like configuration which progressively tilts to 90 degrees at the location of the deltopectoral crest (Asterisk). For
*Figure 8 continued on next page*

*Figure 8 continued*

that reason, the longitudinal trabeculae appear green and the transverse trabeculae appear purple at this location. (B) Dorsal view. (a) Longitudinal virtual thin section of the proximal metaphysis (40 µm thick), (b) transverse virtual thin section in the metaphysis (40 µm thick), (c) detail of (b) showing marrow processes and cavities in transverse section. Abbreviations: c., cavity; dia., diaphysis; Dist., distal end; meta., metaphysis; m.f., mineralisation front; m.p., marrow process; Prox., proximal end; s., sediment; t., trabeculae.

The online version of this article includes the following figure supplement(s) for figure 8:

**Figure supplement 1.** Humeral microanatomical architecture of the stem amniote *Seymouria sanjuanensis* (CM 28597) and the tortoise *Centrochelys sulcata*.

globular calcified cartilage to balance their buoyancy or in extremely retarded developmental conditions, such as paedomorphosis (e.g. pachy-osteosclerotic amniotes, *De Buffrénil et al., 2008*; *De Ricqlès and De Buffrénil, 2001*; crocodiles, *Haines, 1938*), (2) diseased amniotes can have osteosclerotic problems which result in the production and retention of *globuli ossei* (*Gussen, 1967*). In such cases, the production of *globuli ossei* is not solely located in the epiphysis and does not play any role in the elongation process of limb bones. Even though these cases reflect derived and/or rare conditions, they show that amniotes keep the ability to produce *globuli ossei* although they do not allocate them to the limb-bone elongation process.

## Discussion on the batrachian limb-bone elongation strategy

It was hypothesised that a large number of *globuli ossei* would be associated with a slow limb-bone endochondral ossification and development (*De Ricqlès, 1972*; *Haines, 1938*). This was based on the observation of globular calcification in small extant amphibians and neotenic aquatic forms, as well as the rarity or even absence of *globuli ossei* in fast growing juvenile mammals and birds (*De Ricqlès, 1972*; *De Ricqlès, 1979*; *Haines, 1942*; *Quilhac et al., 2014*). The observations contained herein clearly show that long-bone developmental dynamics does not seem to be the leading or unique factor for performing one or the other of the elongation and calcification processes. Indeed, the stem tetrapods *Hyneria*, *Eusthenopteron* and *Acanthostega* (with a humerus remaining cartilaginous for several years), all exhibit the characteristics of a slow appendicular development (and slow somatic development as a whole for *Eusthenopteron* and *Acanthostega* [*Sanchez et al., 2014*; *Sanchez et al., 2016*]) but only produce very few *globuli ossei* (*Kamska et al., 2018*; *Sanchez et al., 2014*; *Sanchez et al., 2016*). On the contrary, they all present an obvious longitudinal metaphyseal spongiosa strongly supporting the development of a hypertrophic columnar cartilaginous growth plate.

The somatic size and ecology were also hypothesised to play a role in limb-bone elongation strategy (*De Ricqlès, 1972*). Once again, our data challenge this hypothesis. Both seymouriamorphs, *Seymouria* and *Discosauriscus*, and the temnospondyl *Metoposaurus* exhibit the same trabecular pattern despite different somatic sizes (skull length of an adult *Seymouria* estimated to 9.5 cm, *Berman et al., 2000*; skull length of a possibly adult *Discosauriscus* estimated to 6.2 cm, *Klembara, 1995*; *Klembara, 2009*; skull length of an adult *Metoposaurus* estimated to 40–50 cm, *Sulej, 2007*) and distinct ecologies (*Seymouria* and *Discosauriscus* being (supposedly) terrestrial, *Berman and Martens, 1993*; *Klembara, 2009*; *Klembara and Meszároš, 1992*; and *Metoposaurus* being aquatic, *Schoch and Milner, 2000*).

*Felisbino and Carvalho, 2001* investigated the limb-bone ossification of the amphibian *Rana*. They observed a late calcification and late ossification of the trabeculae in *Rana catesbeiana* which did not contribute to the bone elongation (*Felisbino and Carvalho, 2001*). The authors therefore suggested that the production of *globuli ossei* could probably play a greater role in reinforcing the limb-bone microstructure – for jumping after a certain age – rather than being associated with its elongation. Because urodeles do not jump despite their late ossification onset, the reasons for them to produce many *globuli ossei* could not be justified as such.

The use of exclusive globular calcification and *globuli ossei* for long-bone endochondral ossification and elongation would therefore probably result from the combination of multiple factors shared by both batrachians and dissorophoids. In order to precisely identify these factors, an extended histological study will have to be carried out within temnospondyls (considering as many environmental factors as possible, including their ecologies and sizes) to draw strong and broad conclusions on this

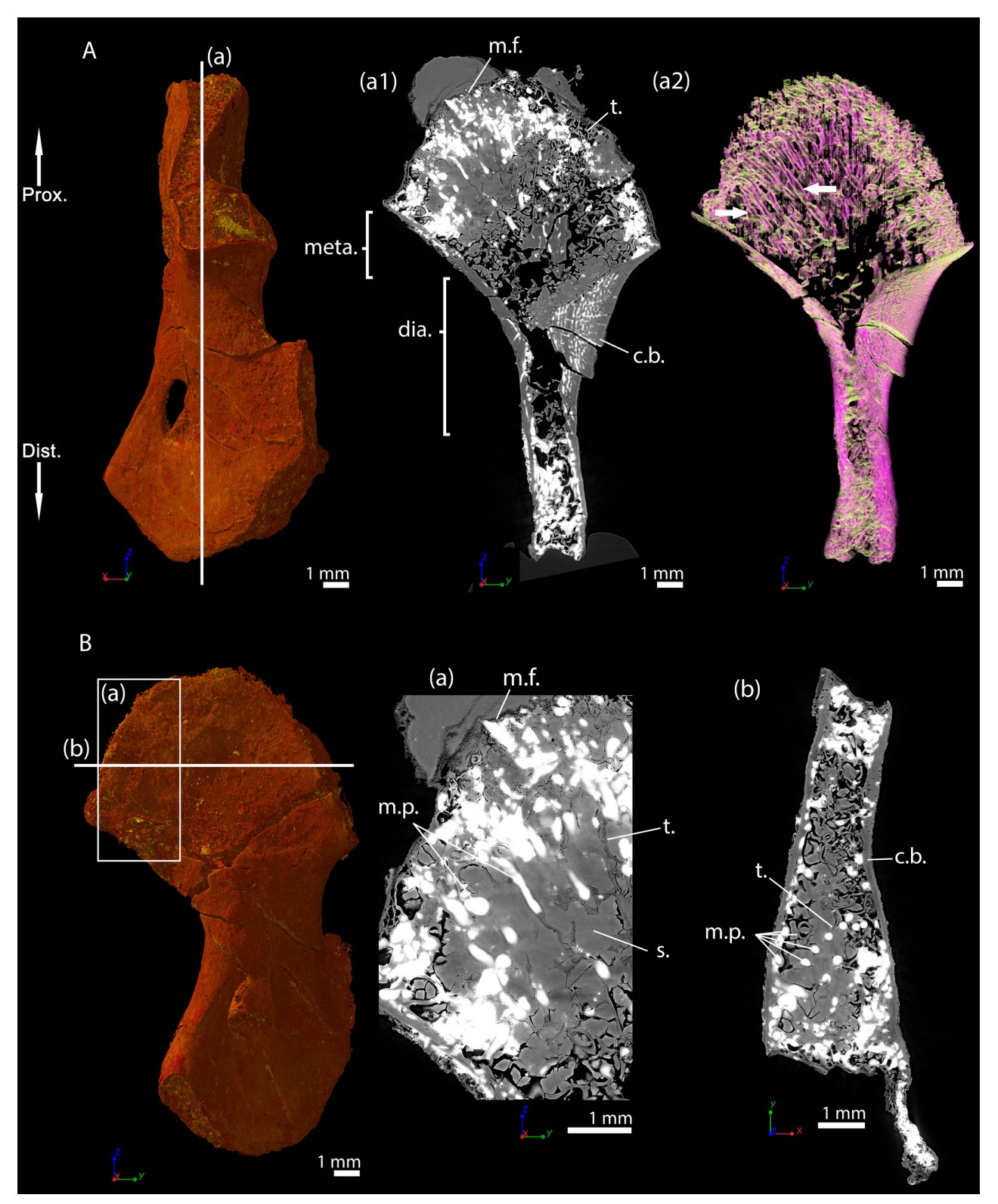

**Figure 9.** Right humerus of a subadult *Discosauriscus austriacus,* SNM Z 15568 imaged using PPC-SRμCT. Due to processing to convert the scan data into a stack of images, the images have been flipped, thereby resulting in a flipped 3D model. (**A**) Frontal view. (**a1**) Longitudinal virtual thin section (40 μm thick) and (**a2**) longitudinal section of the segmented model of the bone (160 μm thick). The longitudinally-oriented trabeculae are highlighted in purple, while the transversally-oriented trabeculae appear in green. (**B**) Ventral view. (**a**) Longitudinal virtual thin section of the proximal metaphysis (40
*Figure 9 continued on next page*

*Figure 9 continued*

µm thick) and (**b**) transverse virtual thin section in the proximal metaphysis (40 µm thick). Abbreviations: c.b., cortical bone; dia., diaphysis; Dist., distal end; meta., metaphysis; m.f., mineralisation front; m.p., marrow process; Prox., proximal end; s., sediment; t., trabeculae.

evolutionary pattern and the reasons for it to be that restricted in the evolutionary history of tetrapods. Nevertheless, the current study shows that amphibians, often considered as models for exhibiting primitive tetrapod features, should be regarded as a clade with a significantly derived evolutionary history, at least with respect to their skeleton.

## Evolution of limb-bone marrow processes

As the long bones of *Apateon* were thin sectioned, it was not possible to assess the 3D organisation of the trabecular mesh nor the calcified cartilage mass to check out the potential presence of marrow processes.

The microanatomy of the other taxa (*Metoposaurus*, *Seymouria*, *Discosauriscus*), however, could be investigated in 3D using PPC-SRµCT. It revealed tubular structures which can be confidently interpreted as marrow processes (on the basis of their shape, size and location in the bone), as seen in extant crocodiles (*Haines, 1938*). The marrow processes in these groups differ from the closed system observed in the stem tetrapod *Eusthenopteron* (*Sanchez et al., 2014*). Instead the marrow processes in *Metoposaurus* open up into multilocular spaces in the trabecular mesh of the metaphysis. In *Discosauriscus* and *Seymouria*, the marrow processes lead to a series of small interconnected cavities (m.c., *Figure 10*) which connect to other tubular processes and open up into the medullary cavity of the bone shaft. The size and shape of these interconnected cavities are variable. These small cavities would therefore rather correspond to a primary regionalisation of the marrow environment – as seen in amniotes like *Crocodylus* (*Haines, 1938*). As a result of intense erosion, the tubular marrow processes in *Discosauriscus* more often directly plug into the medullary cavity of the shaft. On the contrary, the shaft of *Metoposaurus* is highly crossed by thick trabeculae (*Table 2*) forming multilocular spaces as in *Andrias* (*Sanchez et al., 2014*; *Figure 10—figure supplement 1*).

The tip of the marrow process forming a tube penetrating the hypertrophic cartilage of the growth plate in crocodiles is located at the level of the mineralisation front, and in mammals, that is at the base of the calcifying layer of hypertrophic cartilaginous cells (*Figure 1A*). The tip of the marrow process plays a role in initiating endochondral ossification with marrow cells releasing lytic enzymes that degrade the calcified cartilaginous matrix (*Suzuki et al., 1981*). In amniotes, the marrow process directly connects with the medullary cavity of the shaft (*Haines, 1942*) where blood vessels supply growth factors to initiate the ossification (*Gerber et al., 1999*). The bone marrow also produces haematopoietic cells and stem cells (HSC) which need to remain in regulated microenvironments called niches (*Zhang et al., 2003*). The latter are located in the metaphysis, in obligatory proximity with both endothelial and endosteal surfaces (*Wilson and Trumpp, 2006*). For that reason, haematopoiesis only occurs in long bones whose shafts are greatly opened (*Bazzini et al., 1986*; *Tanaka, 1976*). In mammals, the medullary cavity of the shaft can be infilled by calcified cartilage or numerous trabeculae (e.g. pachyostotic condition, *De Ricqlès and De Buffrénil, 2001*). A study was conducted on amedullar, pachyostotic long bones of manatees (*Trichechus manatus*, *Bazzini et al., 1986*). Because haematopoiesis cannot be hosted in their long-bone medullary shaft, manatees have evolved an alternative primary site of haematopoiesis in their vertebral bodies (*Bazzini et al., 1986*). In amphibians, bays of erosion progressively form within the cartilage of the medullary cavity during the development. They can either be isolated forming multilocular spaces separated by bony septa (sept., *Figure 1B1*) or forming an open medullary cavity (*Figure 1B2*). Marrow processes in the humerus of the aquatic giant salamander *Andrias* (*Figure 10—figure supplement 1*) show interconnected tubular structures which open up onto spaces separated by septa in the shaft (*Figure 10—figure supplement 1*). Long bones are probably deprived of haematopoietic activity in this taxon as initial sites of haematopoieisis are located in the liver (*Akiyoshi and Inoue, 2012*). No haematopoietic activity could be observed either in the long bones of the urodele *Triturus (Cynops) pyrrhogaster* exhibiting a multilocular configuration (*Tanaka, 1976*; *Figure 1B1*). However, amphibians with open long-bone medullary cavities (*Figure 1B2*) produce blood cells (e.g. *Rana catesbeiana*, *Tanaka, 1976*) and exhibit no haematopoietic liver structure (e.g. *Akiyoshi and Inoue, 2012*). In the

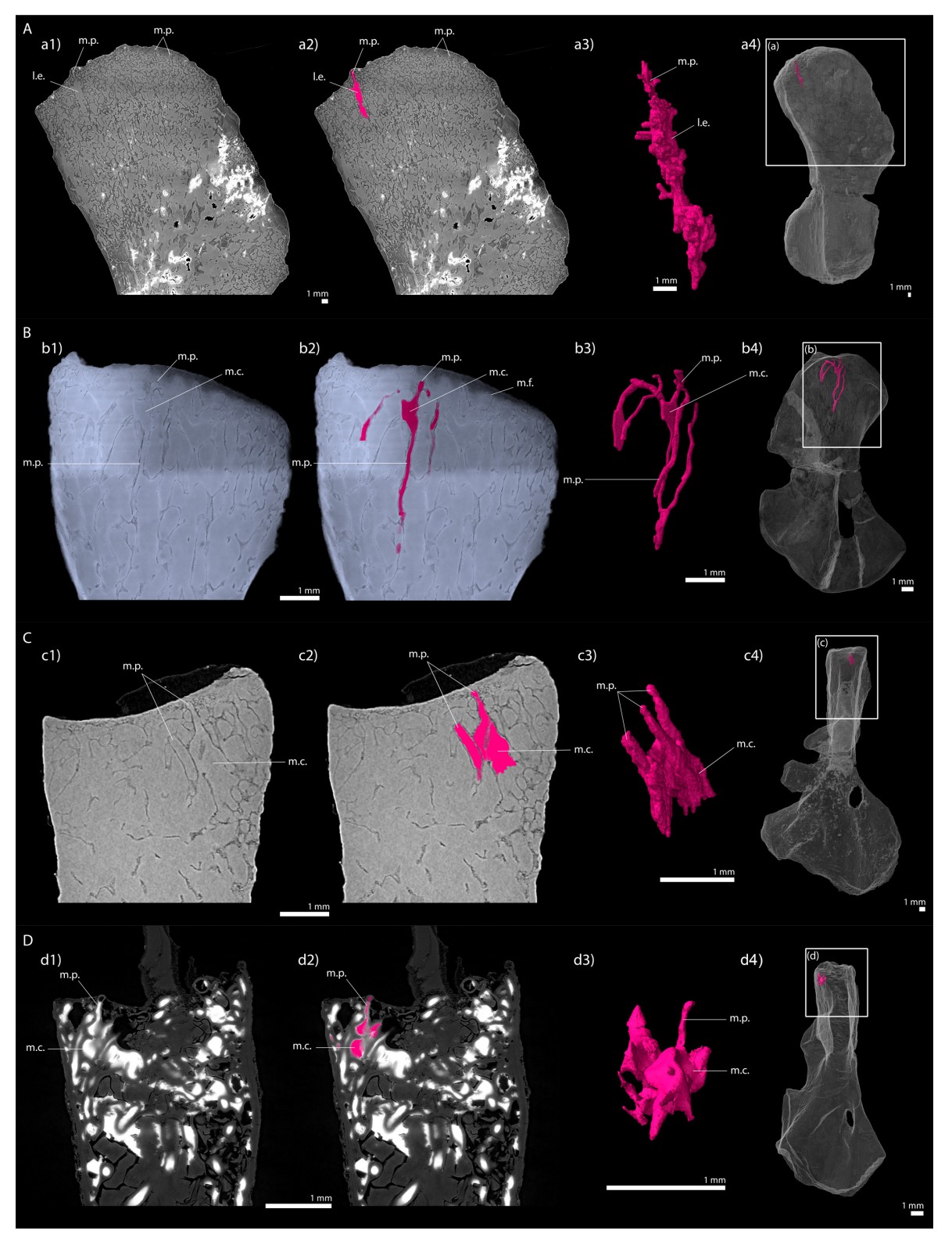

**Figure 10.** Longitudinal virtual sections and three-dimensional (3D) segmentation from PPC-SRµCT of marrow processes and marrow cavities in the humeral proximal ends of: A, *Metoposaurus* sp. (MUZ PGI OS-220/171); B, *Seymouria sanjuanensis* (MNG 7747); C, *S. sanjuanensis* (CM 28597); D, *Discosauriscus austriacus* (SNM Z 15568). (a1, b1, c1, d1) Longitudinal virtual thin section (60 µm thick); (a2, b2, c2, d2) marrow processes and cavities segmented; (a3, b3, c3, d3) 3D models of the segmentations. Note that the marrow cavities have not been completely segmented in 3D to allow the

*Figure 10 continued on next page*

*Figure 10 continued*

full visualisation of the marrow processes; (a4, b4, c4, d4) respective locations of a3, b3, c3, d3 in the humeri. Abbreviations: l.e., region of local erosion; m.c., marrow cavity; m.f., mineralisation front; m.p., marrow process.

The online version of this article includes the following figure supplement(s) for figure 10:

**Figure supplement 1.** Longitudinal virtual sections and three-dimensional (3D) segmentation from PPC-SRµCT of marrow processes in the humeral proximal end of (**A**) *Andrias* sp.

case of an open medullary cavity, marrow vessels run from a central vein into accessory sinusoids (s., *Figure 1B2*) to form an adequate environment for haematopoietic activity (*Tanaka, 1976*). The centralisation of the vascular and marrow systems is therefore crucial for haematopoiesis to occur in long bones (*Tanaka, 1976*).

Based on these observations, the full compartmentalisation of the marrow processes in *Eusthenopteron* (*Sanchez et al., 2014*), as well as the multilocular arrangement in *Metoposaurus* (*Figure 6*), would probably prevent the formation of a centralised vascular network (as observed in *Andrias*; *Figure 10—figure supplement 1* or *Triturus (Cynops) pyrrhogaster*, *Tanaka, 1976*; *Figure 1B1*). This would eventually deprive the marrow cells from HSC niches. We therefore propose that the marrow processes in *Eusthenopteron* and *Metoposaurus* may have only been involved in the induction of endochondral ossification for the elongation of the fin/limb bone but not in haematopoiesis. The humerus of the Devonian limbed stem tetrapod, *Acanthostega*, also exhibits tubular structures under the mineralisation front of the growth plate (*Sanchez et al., 2016*). They can be identified as marrow processes. They open up onto multilocular spaces separated by numerous septa as in *Metoposaurus* and *Andrias* (*Figure 10—figure supplement 1*). For that reason, it is likely that the vascularisation in the medullary bone of *Acanthostega* was not centralised and no marrow haematopoiesis could be produced in their long bones. The interconnected small cavities opening up into a large medullary cavity as seen in the terrestrial Permian seymouriamorphs *Seymouria* and *Discosauriscus* (*Figure 10B–D*) would therefore presumably constitute one of the first forms of microenvironment for HSC niches. The multiple functions of bone marrow would have been acquired at different times in the history of tetrapod evolution. Bone-marrow initiation of endochondral ossification already existed in finned stem tetrapods while trabecular opening/erosion for haematopoiesis could only be evidenced in the (300-million-year-old) Permian seymouriamorphs so far. The migration of blood-cell production in long bones would therefore not seem to be an exaptation predating the water-to-land transition. We intend to investigate the long-bone microanatomy of early tetrapods to identify the timing of this major evolutionary step and elucidate the question whether haematopoiesis migrated into bone marrow in the first tetrapods who ventured on land (with body fossil evidence from 360 million years ago) or afterwards when the process of terrestrialisation was a bit more advanced during the Carboniferous (350–300 million years ago). This will help clarify the convergent factors – environmental conditions (with temperature changes – *Weiss and Wislocki, 1956*; UV dose – *Kapp et al., 2018*) and/or biological factors (e.g. active locomotion – *Tanaka, 1976*) – accompanying the migration of bone-marrow haematopoietic activity into long bones in both amphibians and amniotes.

## Materials and methods

### Materials

We focus on studying the limb-bone growth plate and marrow processes of the temnospondyls *A. pedestris*, *A. caducus*, and *Metoposaurus* sp., considered as stem amphibians (or at least stem batrachians – including anurans and urodeles) by most authors (e.g. *Anderson, 2008*; *Milner, 1988*; *Pardo et al., 2017*; *Ruta and Coates, 2007*; *Schoch, 2019*; *Schoch and Milner, 2004*; *Sigurdsen and Green, 2011*; *Trueb and Cloutier, 1991*) – although we are aware that some authors have proposed diverging hypotheses (e.g. *Marjanović and Laurin, 2013*; *Vallin and Laurin, 2004*). We also investigate the bone histology of the seymouriamorphs *S. sanjuanensis* and *D. austriacus*, which we consider stem amniotes following general consensus (e.g. *Anderson, 2007*; *Klembara et al., 2014*; *Ruta and Coates, 2007*).

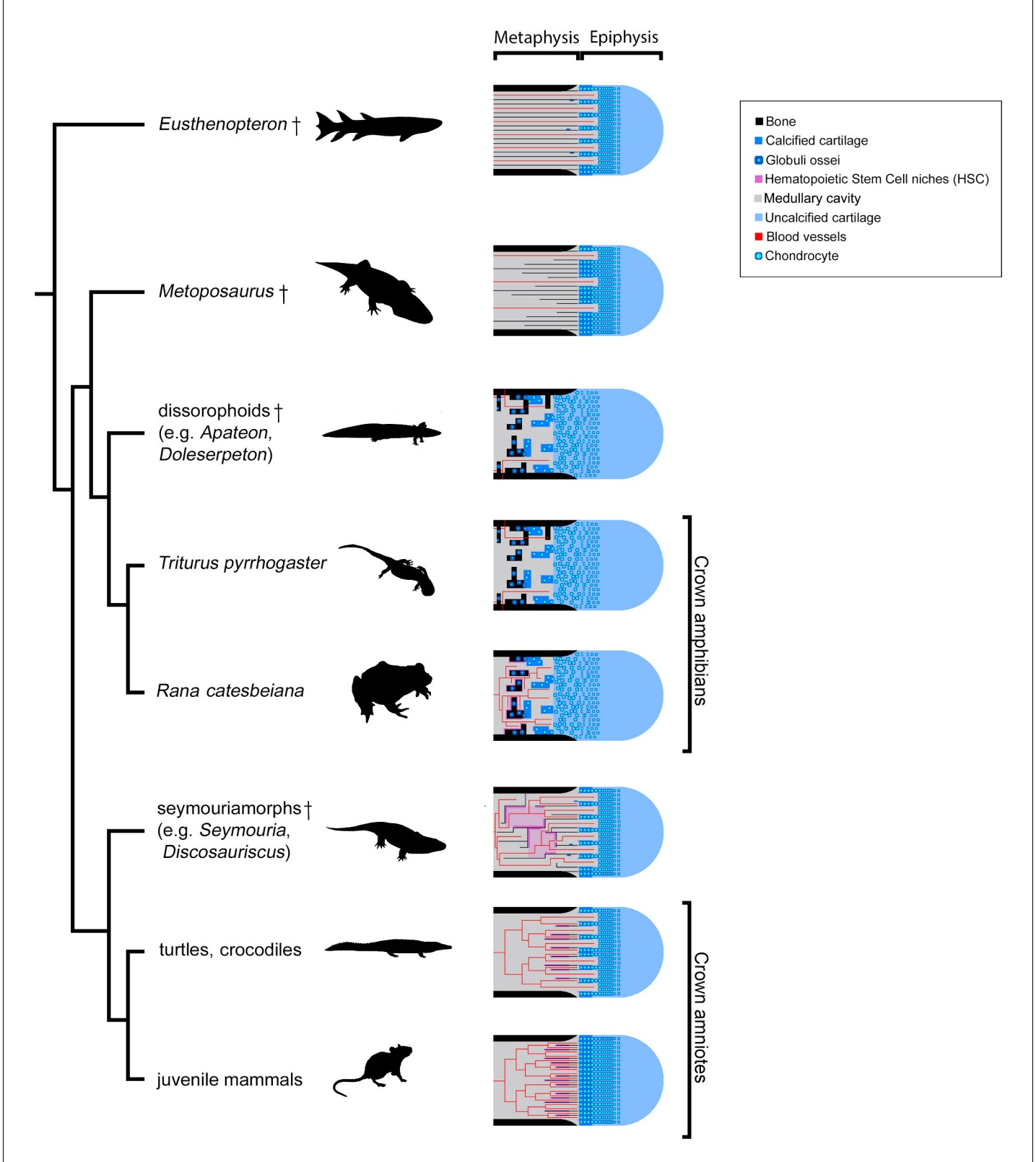

**Figure 11.** Evolution of growth-plate patterns and metaphyseal organisations of the long bones of the studied taxa in a phylogenetic context (e.g., *Ruta and Coates, 2007*; *Schoch, 2019*). Hypothesis on haematopoietic activity is herein contextualised. Black silhouettes represent the taxa studied. Crosses (†) have been attributed to fossil taxa.

The specimens of *A. caducus* (GPIM-N 1297 and GPIM-N 1572) were discovered in the Carboniferous-Permian locality of Erdesbach, Saar Nahe Basin, Germany. GPIM-N 1297 is a juvenile individual (*Figure 2A*) and GPIM-N 1572 is an adult (*Figure 3A*). Their radius and ulna, in addition to the humerus of GPIM-N 1297 exclusively, were longitudinally sectioned through their epiphyses (*Sanchez et al., 2010a*). The specimens of *A. pedestris* (SMNS 54981 – *Figure 4A* – and SMNS 54988 – *Figure 5A*) come from the Carboniferous-Permian locality of Odernheim, Saar Nahe Basin, Germany (*Boy, 2003*; *Sanchez et al., 2010a*; *Sanchez et al., 2010b*). They are both adult specimens (*Sanchez et al., 2010a*). Longitudinal thin sections were taken from their humerus, radius and ulna (*Sanchez et al., 2010a*). All specimens of *Apateon* were found unambiguously as articulated fossil skeletons. The specimens of *Apateon* referred to as SMNS and GPIM-N belong to the collections of the Staatliches Museum für Naturkunde (Stuttgart, Germany) and the specimens marked with MB. Am to the Museum für Naturkunde (Berlin, Germany).

The humerus of *Metoposaurus* sp. comes from the Late Triassic locality of Skarszyny in southern Poland. This was found as an isolated bone in association with small bone fragments and calcified plant remains. The specimen could be assigned to the genus of *Metoposaurus* (Temnospondyli) on the basis of the following morphological characters: (1) a short and slender shaft, (2) a wide head, (3) a pronounced ectepicondyle, (4) the base of the radial condyle forms a small prominence on its ventral side and (5) the radial crest continues from the ectepicondyle to the proximal head (*Sulej, 2007*). The size of the humerus suggests that it is from a (sub-)adult individual. The specimen is hosted in the collections of the Geological Museum, Polish Geological Institute – National Research Institute (Warsaw, Poland). The specimen is registered as MUZ PGI OS-220/171 (*Figure 6*).

The material of *S. sanjuanensis* studied here consists of two three-dimensionally preserved humeri: one small and one large individual of *S. sanjuanensis* (respectively, MNG 7747 – *Figure 7* – and CM 28597 – *Figure 8*). MNG 7747 belongs to a specimen found in the Tambach formation, Bromacker locality, Lower Permian of Central Germany (*Berman et al., 2000*; *Berman and Martens, 1993*; *Klembara et al., 2001*) and CM 28597 was excavated in the Cutler Formation, Lower Permian of North-Central New Mexico, USA (*Berman et al., 1987a*; *Vaughn, 1966*). Both bones were found in articulation to the rest of their limbs (MNG 7747 – *Klembara et al., 2001*; CM 28597 – *Berman et al., 1987b*). They were therefore identified unambiguously as the humeri of *S. sanjuanensis* (MNG 7747 – *Berman and Martens, 1993*; *Klembara et al., 2001*; CM 28597 – *Berman et al., 1987b*). MNG 7747 is a small left humerus (*Figure 7*, 17.21 mm long; *Klembara et al., 2001*) of a subadult individual (*Klembara et al., 2001*; *Klembara et al., 2006*). CM 28597 is more than twice larger (*Figure 8*, 39.60 mm long) and belongs to a right limb. It has been designated as an adult specimen based on its skull length (*Estefa et al., 2020*). The specimen MNG 7747 comes from the collections of the Museum der Natur (Gotha, Germany) and CM 28597 from the Carnegie Museum of Natural History (Pittsburgh, USA).

The humerus of *D. austriacus* originates from the Lower Permian of Kochov-Horka in the Czech Republic. The specimen of *D. austriacus*, SNM Z 15568, is stored in the collections of the Slovak National Museum in Bratislava (Bratislava, Slovakia; *Figure 9*). This bone is the right humerus. It is 18.3 mm long and was found in association with the rest of the body. It belongs to a subadult individual (skull length: 5.2 cm; *Klembara and Bartík, 1999*).

None of these last three specimens were physically sectioned, they were studied using PPC-SRµCT, following the protocols developed for virtual bone histology (*Sanchez et al., 2012*; see Methods section).

## Methods
### Thin sectioning and microscopic observations
The limbs of *A. pedestris* and *A. caducus* were removed from their slabs and embedded in resin (*Sanchez et al., 2010a*; *Sanchez et al., 2010b*). The blocks were sawed with a diamond powder disk, and polished following the protocol published by *Lamm, 2013*. All thin sections were observed and photographed using an optical microscope under natural light (Nikon Eclipse 80i mounted with a Nikon D300 digital camera).

## Propagation phase-contrast X-ray synchrotron radiation micro-computed tomography and data reconstruction

The specimens were imaged at the European Synchrotron Radiation Facility (France; beamline ID19) using Propagation Phase-Contrast X-ray Synchrotron Radiation Micro-Computed Tomography (PPC-SRμCT) (*Tafforeau et al., 2006*). Technical details for the scan data of *D. austriacus* and *S. sanjuanensis* were provided in a former publication (*Estefa et al., 2020*). Here is the protocol used for scanning the humerus of *Metoposaurus* sp. It was imaged with a voxel size of 11.79 μm using a PCO Edge 4.2 camera mounted on a tandem optic composed of Hasselblad 100 mm and Canon 50 mm objectives coupled to a 500 μm thick LuAG:$_{Ce}$ scintillator. The white beam produced by the ID19 wiggler set at a gap of 52 mm was filtered with 1.4 mm of diamond and 12 mm of copper. The resulting detected average energy was about 123 keV. The sample was imaged in half acquisition mode, rotating over 360 degrees with the centre of rotation on the right side of the field of view to enlarge the lateral field of view. 5000 projections of 48 ms each (resulting from an accumulation of 4 sub-frames of 12 ms each) were taken over 360 degrees. The fossil was placed at a propagation distance of 13 m from the detector in order to maximise the phase-contrast effect.

Tomographic slices were reconstructed using filtered back-projection algorithm using the software PyHST2 (*Mirone et al., 2014*) coupled with modified single distance phase retrieval (*Paganin et al., 2002*; *Sanchez et al., 2012*). The different sub-volumes were ring-corrected (*Lyckegaard et al., 2011*) and vertically concatenated, converted in 16 bits and cropped using matlab inhouse developed systems. In order to ease the segmentation of the trabeculae, a tomographic texture enhancement filter was applied (*Cau et al., 2017*; *Figure 12*).

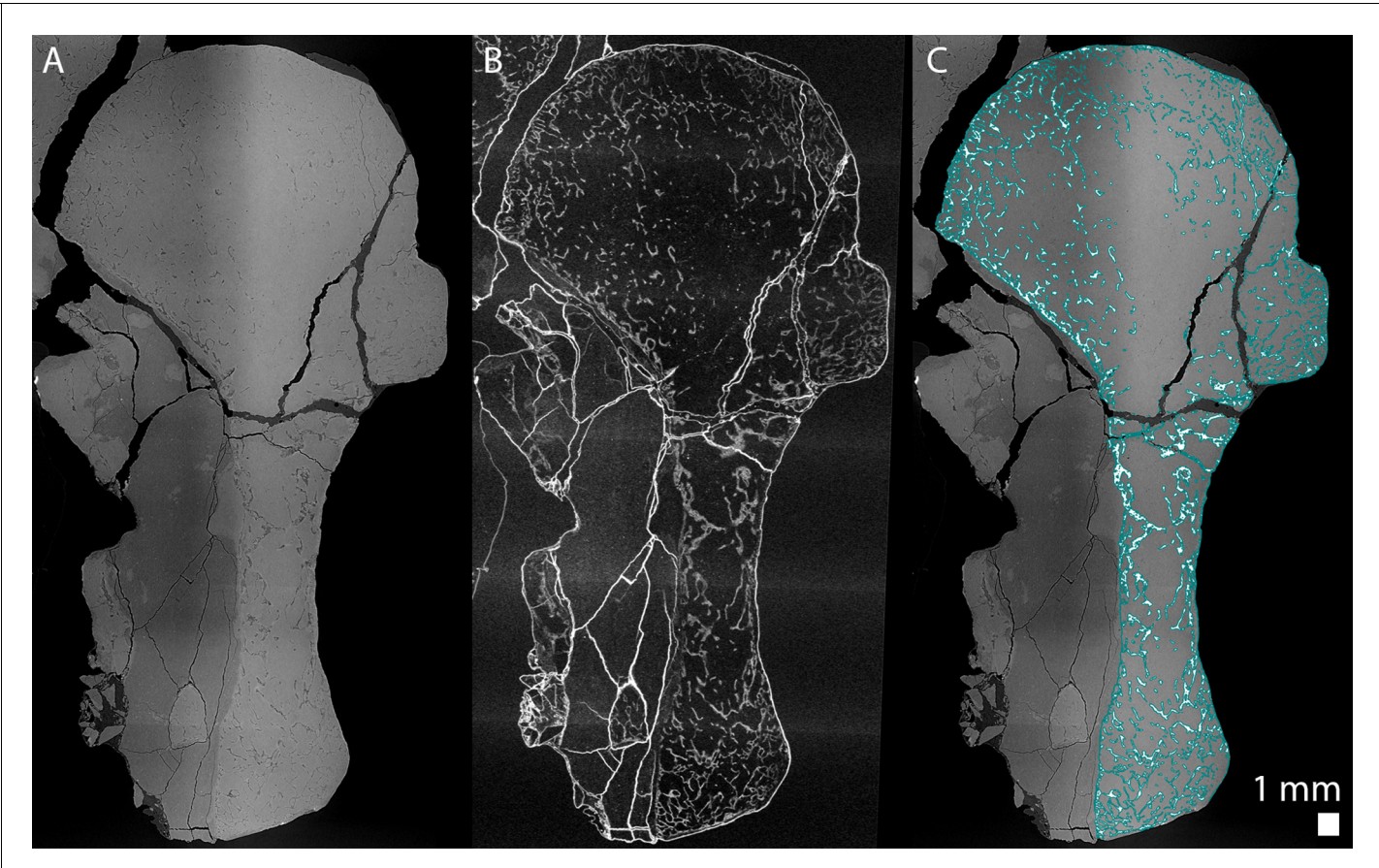

**Figure 12.** Longitudinal virtual thin sections of the right humerus of an adult *Seymouria sanjuanensis* (CM 28597). (**A**) Tomogram showing low frequency artefacts resulting in an image divided into a darker and brighter part, (**B**) image processed with a filter for tomographic texture enhancement to remove low frequency artefacts (*Cau et al., 2017*) and (**C**) overlap of A and the bone segmentation with the aid of B used to produce *Figure 8Aa2*. The segmentation is highlighted by the blue line in C.

## Virtual bone histology

For each sample, virtual thin sections and 3D models were made using *VGStudio MAX* (version 3.2, Volume Graphics Inc, Germany). We used a directional coloured light system to highlight the general orientation of the trabeculae (*Sanchez et al., 2014*). The longitudinal trabeculae appear in purple and the transverse ones in green.

## Measurements

Measurements in the humeri (i.e. thickness of the trabeculae and diameter of the marrow processes) were made with *VGStudio MAX* (version 3.2, Volume Graphics Inc, Germany) (*Estefa et al., 2020*).

## Acknowledgements

The authors thank F Witzmann from the Museum für Naturkunde (Berlin) and R Schoch from the Staatliches Museum für Naturkunde (Stuttgart) for allowing them to section fossils from their collections. We are grateful to B Le Dimet and M Lemoine (MNHN, Paris) for casting the bones and preparing the thin sections of *Apateon*. The authors acknowledge the European Synchrotron Radiation Facility (France) where the samples were imaged (proposals EC203, LS2832, SS). We show our gratitude to P Ahlberg and T Haitina (Uppsala University, Sweden), A de Ricqlès and J-S Steyer (MNHN, France) and R Schoch (SMNS, Germany) for fruitful discussions on the development of temnospondyl limbs, tetrapod growth plate and bone marrow. SS was funded by the European program Synthesys for collecting the samples in museums. This research was supported by two grants from the Vetenskapsrådet (2015–04335, 2019–04595, SS) and the Scientific Grant Agency of Ministry of Education of Slovak Republic and Slovak Academy of Sciences (1/0228/19, JK). AMC acknowledges support from the Australia Research Council (DP 160102460).

## Additional information

### Funding

| Funder | Grant reference number | Author |
|---|---|---|
| Vetenskapsrådet | 2015-04335 | Sophie Sanchez |
| Synthesys | | Sophie Sanchez |
| European Synchrotron Radiation Facility | EC203 | Sophie Sanchez |
| European Synchrotron Radiation Facility | LS2832 | Sophie Sanchez |
| Scientific Grant Agency of Ministry of Education of Slovak Republic and Slovak Academy of Sciences | 1/0228/19 | Jozef Klembara |
| Australian Research Council | DP 160102460 | Alice M Clement |
| Vetenskapsrådet | 2019-04595 | Sophie Sanchez |

The funders had no role in study design, data collection and interpretation, or the decision to submit the work for publication.

### Author contributions

Jordi Estefa, Formal analysis, Investigation, Visualization, Methodology; Paul Tafforeau, Resources, Software, Investigation, Methodology; Alice M Clement, Funding acquisition, Investigation, Methodology; Jozef Klembara, Grzegorz Niedźwiedzki, Funding acquisition, Investigation; Camille Berruyer, Investigation, Data acquisition; Sophie Sanchez, Conceptualization, Resources, Formal analysis, Supervision, Funding acquisition, Investigation, Methodology

## Author ORCIDs

Paul Tafforeau (iD) http://orcid.org/0000-0002-5962-1683
Alice M Clement (iD) http://orcid.org/0000-0003-0380-7347
Grzegorz Niedźwiedzki (iD) https://orcid.org/0000-0002-4775-5254
Camille Berruyer (iD) http://orcid.org/0000-0002-0046-2972
Sophie Sanchez (iD) https://orcid.org/0000-0002-3611-6836

## Decision letter and Author response

Decision letter https://doi.org/10.7554/eLife.51581.sa1
Author response https://doi.org/10.7554/eLife.51581.sa2

## Additional files

### Supplementary files

• Transparent reporting form

### Data availability

The regions of interest of the reconstructed data are available on http://paleo.esrf.eu/. The raw data have been deposited to the European Synchrotron Radiation Facility. All the fossils and thin sections observed are available in the museum/university collections cited in the Materials and Methods section.

The following datasets were generated:

| Author(s) | Year | Dataset title | Dataset URL | Database and Identifier |
|---|---|---|---|---|
| Sanchez S, Estefa J, Tafforeau P | 2021 | Radiographic data of the complete scan of Discosauriscus humerus SNMZ 15568 and associated partial 3D reconstructions | https://doi.esrf.fr/10.15151/ESRF-DC-385889323 | European Synchrotron Radiation Facility, 10.15151/ESRF-DC-385889323 |
| Sanchez S, Estefa J, Tafforeau P | 2021 | Radiographic data of the complete scan of Seymouria juvenile humerus MNG7747 and associated partial 3D reconstructions | https://doi.esrf.fr/10.15151/ESRF-DC-386797910 | European Synchrotron Radiation Facility, 10.15151/ESRF-DC-386797910 |
| Sanchez S, Estefa J, Tafforeau P | 2021 | Radiographic data of the complete scan of Seymouria adult humerus CM-28597 and associated partial 3D reconstructions | https://doi.esrf.fr/10.15151/ESRF-DC-386801611 | European Synchrotron Radiation Facility, 10.15151/ESRF-DC-386801611 |
| Sanchez S, Estefa J, Tafforeau P | 2021 | Radiographic data of the complete scan of Metoposaurus humerus MUZ-PGI-OS-220171 and associated partial 3D reconstructions | https://doi.esrf.fr/10.15151/ESRF-DC-386788297 | European Synchrotron Radiation Facility, 10.15151/ESRF-DC-386788297 |

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
