## [Decision Letter]

**Acceptance summary:**

Sanchez et al. investigated the long bones in fossils of several stem amphibians and amniotes. In particular they focused on strategies of long-bone elongation and discussed the role of the globuli ossei in this process. The authors suggested that a large medullary cavity coupled with haematopoiesis occurs first in Permian tetrapods. This adds the fossil dimension to the currently active discussion on the evolution of the bone marrow niche.

**Decision letter after peer review:**

Thank you for submitting your article "New light shed on the early evolution of limb-bone growth plate and bone marrow" for consideration by *eLife*. Your article has been reviewed by two peer reviewers, and the evaluation has been overseen by Diethard Tautz as the Senior and Reviewing Editor. The following individuals involved in review of your submission have agreed to reveal their identity: Michael Blumer (Reviewer #1); Holly Woodward (Reviewer #2).

The reviewers have discussed the reviews with one another and the Reviewing Editor has drafted this decision to help you prepare a revised submission.

This manuscript addresses the evolution of haematopoiesis within tetrapods and its occurrence in red bone marrow, and attempts to address hypotheses regarding the location of RBC production within long bones and the water-to-land transition. The authors conclude that based on their 3D imaging and histological analyses, the migration of RBC production into long bones is not an exaptation predating the water-to-land transition and instead originated with the earliest tetrapods. The study is well-performed and well-written. However, there are some minor points of criticism that need the attention of the authors.

Reviewer #1:

1) Introduction, second paragraph: Do the authors refer here to amphibians or amniotes? In long bones of chicken and mice, for instance, endochondral bone development of the diaphysis (= formation of the primary ossification center) starts with mineralization of the cartilage matrix followed by vascular invasion coupled with cartilage resorption and new formation of bone matrix. Consequently mineralization of the cartilage matrix occurs first followed by erosion of this matrix.

2) Introduction, third paragraph: The authors should also mention that VEGF stimulates the ingrowth of blood vessels into the diaphysis, an important step for endochondral bone formation.

3) Introduction, third paragraph: Are there really mature animals where a cartilaginous growth plate is retained? It's only a question because I am not aware of this fact.

4) Results, subsection “*Apateon caducus*, juvenile specimen GPIM-N 1297, Radius and Ulna”: Why is written epiphysis? Figures 2Cc1, 3 show only a higher magnification of the metaphysis (the brackets in Figure 2C clearly indicate the region of the metaphysis) that contains globular ossei.

5) Results, subsection “*Apateon pedestris*, adult specimen SMNS 54981, Humerus”: e.l. (erosion lacunae) is not shown in Figure 4B.

6) Results, subsection “*Apateon pedestris*, adult specimen SMNS 54988, Humerus”: it should read: g.o., Figure 5Bb.

7)Results, subsection “*Metoposaurus* sp., (sub-)adult specimen ZPAL (without number), Humerus”: it should read: Figure 6Aa1-2.

8) Figure 6Bc is neither mentioned in the text nor in the legends to the figure. Furthermore, from the image it is not clear which (mineralized) tissue displays the Liesegang's rings? Only in the Discussion I found a short reference to Figure 6Bc that, however, doesn't shed light on the figure.

9) Figure 8Aa2 (yellow arrow) and Results, subsection “*Seymouria sanjuanensis*, adult specimen CM 28597, Humerus”: Based on which histological observation do the authors conclude that there is "arrested growth" in this area of the humerus?

10) Results, subsection “*Discosauriscus austriacus*, subadult specimen SNM Z 15568, Humerus”, last sentence: it should read "was covered".

11) Figure 11. The regions of the long bone (diaphysis, metaphysis and epiphysis) are not denoted correctly. Compare with Figure 1.

Discussion: I do not understand why the authors put in brackets "and somatic development – as a whole": Maybe the brackets are not correctly placed.

Reviewer #2:

Within the Introduction, could the authors provide one or two sentences to explain where/how red blood cell production occurs within modern ray-finned fish, to compare with the marrow RBC production in tetrapods?

The sentence in the subsection “Discussion on the batrachian limb-bone elongation strategy” seems to be missing a word after "appendicular".

---

## [Author Response]

Reviewer #1:1) Introduction, second paragraph: Do the authors refer here to amphibians or amniotes? In long bones of chicken and mice, for instance, endochondral bone development of the diaphysis (= formation of the primary ossification center) starts with mineralization of the cartilage matrix followed by vascular invasion coupled with cartilage resorption and new formation of bone matrix. Consequently mineralization of the cartilage matrix occurs first followed by erosion of this matrix.

This paragraph actually is a general paragraph. We agree with reviewer 1 that the process of endochondral ossification can differ in some aspects between amphibians and amniotes. For this reason, we have decided to rewrite our general definition of endochondral ossification as follows:

“In this region, the cartilage is progressively substituted with bone: this process is called endochondral ossification (Francillon-Vieillot et al., 1990; Hall, 2005).”.

Also, we have decided to provide more details in the paragraph relating to amphibians to highlight the differences in endochondral ossification between amniotes and amphibians. In 1964, de Ricqlès followed the long-bone histology of *Pleurodeles waltl*. He found out that endochondral ossification started at a later stage than in mammals. He first observed a reduction of the cartilaginous matrix with the formation of lacunae, that were subsequently filled in with bone marrow, long before endochondral ossification starts. The first sentence of our paragraph clearly states that the process was different between amphibians and amniotes. We emphasized this aspect with this additional description:

“Unlike mammals, endochondral ossification starts at a later stage in urodeles (De Ricqlès, 1964). The diaphyseal cartilaginous matrix is first hollowed by the formation of lacunae that are subsequently filled in with bone marrow, far before endochondral ossification starts (De Ricqlès, 1964).”

2) Introduction, third paragraph: The authors should also mention that VEGF stimulates the ingrowth of blood vessels into the diaphysis, an important step for endochondral bone formation.

Following the advice of reviewer 1, we added this information to the paragraph:

“Growth factors, such as the vascular endothelial growth factor (VEGF), trigger cartilage calcification and regulate endochondral ossification through stimulation of blood-vessel ingrowth into the diaphysis (Gerber et al., 1999).”

In amphibians, it seems to be less studied. A peak of VEGF expression is present in the hindlimb at the metamorphic climax in the amphibian *Bufo gargarizan*, but absent during the pre-metamorphic period (Gao et al., 2018). Diaphyseal periosteal ossification has already started at this stage (Gao et al., 2018). Bo et al. observed an increase of endochondral ossification activity during metamorphosis (Bo et al., 2018). We therefore know that VEGF plays a major role in endochondral ossification in amphibians but it seems to play a role at a later stage, when endochondral ossification has reached a certain degree. As no further details are given in the literature, we decided to retain our sentence on the role of VEGF as follows:

“In mammals and birds (e.g., mouse, Zelzer et al., 2002; chicken, Carlevaro et al., 2000), VEGF initiates vascular ingrowth before endochondral ossification starts. […] VEGF would therefore seem to play a major role in amphibian long-bone endochondral ossification as well, but, this role still needs to be characterized.”

3) Introduction, third paragraph: Are there really mature animals where a cartilaginous growth plate is retained? It's only a question because I am not aware of this fact.

Yes, it has been demonstrated that the growth of long bones ceases before the growth plate fuses in some animals that are considered sexually and skeletally mature. However, this only happens in some taxa and the reasons for it are still being investigated.

Here are the articles in which we found this information:

Parfitt, 2002: “the data demonstrate that epiphyseal fusion does not precede, but rather follows, the cessation of growth. […] Eventually, it is resorbed completely to establish continuity between the cancellous bone of the epiphysis and metaphysis, although it may persist radiographically for several years as a so-called epiphyseal scar.”

Roach et al., 2003: “In larger mammals (rabbit upwards), the growth plates close at skeletal maturity and longitudinal growth ceases. Smaller rodents (rats, mice), however, maintain a growth plate into old age (Mehta et al. in press). […] The reason for the continued presence of a growth plate in the absence of longitudinal growth is not understood, nor is it known why linear growth no longer occurs.”

4) Results, subsection “*Apateon caducus*, juvenile specimen GPIM-N 1297, Radius and Ulna”: Why is written epiphysis? Figures 2Cc1, 3 show only a higher magnification of the metaphysis (the brackets in Figure 2C clearly indicate the region of the metaphysis) that contains globular ossei.

We agree with reviewer 1 and deleted “epiphysis” from the sentence.

5) Results, subsection “*Apateon pedestris*, adult specimen SMNS 54981, Humerus”: e.l. (erosion lacunae) is not shown in Figure 4B.

Sorry, it was labeled as e.b. on Figure 4B. We have updated the text accordingly.

6) Results, subsection “*Apateon pedestris*, adult specimen SMNS 54988, Humerus”: it should read: g.o., Figure 5Bb.

We agree. We have updated the text accordingly.

7) Results, subsection “*Metoposaurus* sp., (sub-)adult specimen ZPAL (without number), Humerus”: it should read: Figure 6Aa1-2.

Thank you to reviewer 1 for checking all this carefully. We have changed the text accordingly.

8) Figure 6Bc is neither mentioned in the text nor in the legends to the figure. Furthermore, from the image it is not clear which (mineralized) tissue displays the Liesegang's rings? Only in the Discussion I found a short reference to Figure 6Bc that, however, doesn't shed light on the figure.

We have corrected that with a sentence in the Results section:

“Some remnants of calcified cartilage are visible through Liesegang’s rings forming within the cartilage remaining between the metaphyseal trabeculae (Figure 6Bc).”

We have also decided to modify Figure 6Bc by zooming it in and relabeling it.

9) Figure 8Aa2 (yellow arrow) and Results, subsection “*Seymouria sanjuanensis*, adult specimen CM 28597, Humerus”: Based on which histological observation do the authors conclude that there is "arrested growth" in this area of the humerus?

Because it was not obvious that the metaphyses in the humerus of *Seymouria* would be crossed by lines of stress on Figure 8, we decided to provide an additional figure (Figure 8—figure supplement 1). This figure illustrates a radiography of the humerus of *Seymouria* and a thick section made in the humerus of a tortoise (*Centrochelys sulcata*) to show similarities with the lines of stress (called Harris lines) in mammals (Harris, 1933; Garn et al., 1968) and birds (Wegner, 1874).

We also added the following paragraph to explain the phenomenon:

“A resting surface (yellow arrow, Figure 8Aa2, red arrow, Figure 8—figure supplement 1A), can be observed 2-to-4 mm under the mineralisation front. […] Harris lines seem to result from both short- and long-term pressures (e.g., starvation – Park, 1964; disease and deficiencies – Duckler and Van Valkenburgh, 1998).”

10) Results, subsection “*Discosauriscus austriacus*, subadult specimen SNM Z 15568, Humerus”, last sentence: it should read "was covered".

Thanks. We corrected this grammar error.

11) Figure 11. The regions of the long bone (diaphysis, metaphysis and epiphysis) are not denoted correctly. Compare which Figure 1.

Thanks. We corrected this error.

Discussion: I do not understand why the authors put in brackets "and somatic development – as a whole": Maybe the brackets are not correctly placed.

The suggestion of a slow somatic development could not be proposed for *Hyneria* because we don’t have a full skeletochronological sequence due to osteoclastic erosion in the humerus. However, skeletochronological data were available for assessing the life history traits of *Eusthenopteron* and *Acanthostega*: *Eusthenopteron* became adult after 11 years of pre-reproductive period (Sanchez et al., 2014) and the juvenile stage of *Acanthostega* was lasting at least 6 years (Sanchez et al., 2016). Based on the comment of reviewer 1, we considered that we needed to clarify this and decided to change our sentence as follows:

“the stem tetrapods *Hyneria*, *Eusthenopteron* and *Acanthostega* (with a humerus remaining cartilaginous for several years), all exhibit the characteristics of a slow appendicular development (and slow somatic development as a whole for *Eusthenopteron* and *Acanthostega*, Sanchez et al., 2014; Sanchez et al., 2016) but only produce very few *globuli ossei* (Kamska et al., 2019; Sanchez et al., 2014; Sanchez et al., 2016).”

Reviewer #2:Within the Introduction, could the authors provide one or two sentences to explain where/how red blood cell production occurs within modern ray-finned fish, to compare with the marrow RBC production in tetrapods?

Yes, we did it:

“After birth, bone marrow is the definitive haematopoietic system in mostly terrestrial mammals and many other tetrapods (Akiyoshi and Inoue, 2012; Kapp et al., 2018; Orkin and Zon, 2008) but not in fish nor some aquatic tetrapods (Akiyoshi and Inoue, 2012; Avagyan and Zon, 2016; Kapp et al., 2018). Indeed, red blood cells are produced in the supraspinal organ in the lamprey, the kidney and liver in actinopterygians (ray-finned fish) and some amphibians (tadpoles and aquatic adults, Akiyoshi and Inoue, 2012), and the kidney in lungfish (Amemiya et al., 2007; Kapp et al., 2018).”

The sentence in the subsection “Discussion on the batrachian limb-bone elongation strategy” seems to be missing a word after "appendicular".

The word “development” was missing indeed so we added it in the sentence:

“Indeed, the stem tetrapods *Hyneria*, *Eusthenopteron* and *Acanthostega* (with a humerus remaining cartilaginous for several years), all exhibit the characteristics of a slow appendicular development (and slow somatic development as a whole for *Eusthenopteron* and *Acanthostega*, Sanchez et al., 2014; Sanchez et al., 2016) but only produce very few *globuli ossei* (Kamska et al., 2019; Sanchez et al., 2014; Sanchez et al., 2016).”

References:

Parfitt AM. 2002. Misconceptions (1): epiphyseal fusion causes cessation of growth. Bone 30:337-339.

Roach HI, Mehta G, Oreffo ROC, Clarke NMP, Cooper C. 2003. Temporal analysis of rat growth plates: cessation of growth with age despite presence of a physis. Journal of Histochemistry and Cytochemistry 51:373-383.